# Dissociable cognitive strategies for sensorimotor learning

Samuel D. McDougle [1] & Jordan A. Taylor[2,3]

Computations underlying cognitive strategies in human motor learning are poorly understood. Here we investigate such strategies in a common sensorimotor transformation task. We show that strategies assume two forms, likely reflecting distinct working memory representations: discrete caching of stimulus-response contingencies, and time-consuming parametric computations. Reaction times and errors suggest that both strategies are employed during learning, and trade off based on task complexity. Experiments using pressured preparation time further support dissociable strategies: In response caching, time pressure elicits multi-modal distributions of movements; during parametric computations, time pressure elicits a shifting distribution of movements between visual targets and distal goals, consistent with analog re-computing of a movement plan. A generalization experiment reveals that discrete and parametric strategies produce, respectively, more localized or more global transfer effects. These results describe how qualitatively distinct cognitive representations are leveraged for motor learning and produce downstream consequences for behavioral flexibility.

[1] Department of Psychology, University of California, Berkeley, 2121 Berkeley Way, Berkeley, CA 94704, USA. [2] Department of Psychology, Princeton University, Peretsman-Scully Hall, Princeton, NJ 08540, USA. [3] Princeton Neuroscience Institute, Princeton University, Peretsman-Scully Hall, Princeton, NJ 08540, USA. Correspondence and requests for materials should be addressed to S.D.M. (email: mcdougle@berkeley.edu)

When first learning a new motor skill, selecting an appropriate action can be a time-consuming, deliberative process. Consider someone first learning to play the piano: ideally, she could quickly learn a stimulus–response mapping relating notes on the staff to their appropriate keys. However, learning this mapping is only tractable when a musical score has a few notes in a small range. As it gets more complicated, things fall apart—if we just consider just the number of keys in an octave, it easily exceeds our typical working memory capacity[1]. A common strategy (used in piano pedagogy) to overcome this limitation is to approach it parametrically: She can anchor her thumb on middle C and reference other notes on the lines of the musical staff relative to this key. While this strategy affords her the ability to play a more complex melody within a few minutes of practice, it also becomes increasingly cumbersome the further a given note is from middle C—echoing the phenomena of mental scanning and mental rotation[2,3]. These two strategies, one a discrete map (caching stimulus–response pairs) and the other a parametric algorithm (computing relative distances from C), offer two ways to approach learning a novel motor skill.

In simpler motor tasks, like the visuomotor rotation task[4], subjects often leverage strategies to rapidly improve performance[5]. Strategic processes appear to be related to higher reaction times[6], improved task performance[7], and the direction of eye gaze[8,9]. Increased reaction time, and the fact that strategies are often verbalizable, suggests that they reflect deliberative, controlled processing[10]. Control processes often rely on working memory, making it one candidate system that may underlie cognitive strategies for motor learning. Evidence suggests that spatial working memory ability correlates with performance in visuomotor tasks[11], and both recruit similar neural circuits[12]. Moreover, spatial working memory ability correlates with the use of explicit strategies in visuomotor rotation learning[13]. However, it remains unclear what kinds of working memory representations are used for strategies in motor learning.

Here we set out to directly characterize cognitive strategies in a visuomotor rotation task, which requires subjects to adapt to sensory feedback that is rotated relative to their movements. We hypothesized that strategies would take two broad forms, either discrete response caching (RC) or parametric mental rotation (MR). RC is here defined as the maintenance of acquired one-to-one associations between a set of stimuli and a set of responses maintained in memory[14,15], perhaps relying on processing in prefrontal cortex[16,17]. As a form of item-based working memory, the efficacy of RC should be subject to load (e.g., the number of items to be stored[14,18]).

A parametric MR strategy is the canonical example of an analog computation[3,19], in which an internal mental representation is manipulated in visual working memory like a physical object[20]. Evidence from behavior and neurophysiology provides support for mental rotation in the planning of reaching movements: reaction time (RT) during the mental rotation of reaches scales with the magnitude of required rotation, sharing remarkable similarities to visual mental rotation[21–23]. Moreover, decoded neuronal population vectors in motor cortex appear to rotate through directional space during mental rotation of a reach plan[24] (but see ref. [25]).

Critically, mental rotation can compute rotations with arbitrary signs or magnitudes, and through multiple planes[3], thus constituting a flexible algorithm that is formally equivalent to the application of a rotation matrix to some mental representation. Similar algorithms likely exist for other sensorimotor transformations: for instance, linear RT effects are observed when humans compute varying gains on reaching extent;[23] here, the heuristic would represent a scalar transformation rather than a

rotation, though the same logic applies. The idea of mental rotation specifically describing explicit learning in visuomotor rotation tasks has been suggested before[7], and here we provide the first direct test of this idea.

A key question concerns how cognitive strategies change over the course of learning. Recent work on visual mental rotation supports the idea of parametric vs. discrete strategies in that domain—if subjects are exposed to many unique objects in a visual mental rotation task, RT signatures of mental rotation persist over days; however, if they are only exposed to a few images during extended training, mental rotation effects diminish with time, suggesting a shift to item-based retrieval[15]. Similarly, recent work on visuomotor rotation learning suggests that time-consuming strategic learning processes appear to become more automatic with practice[26]. Together, these findings are broadly consistent with Logan's theory of skill automatization[27], where learning proceeds from an algorithmic stage to an associative stage, the latter requiring repeated practice of specific instances. Our experiments here are poised to confirm this transition as a model of the cognitive stages of visuomotor learning, directly characterize the computations underlying these different stages, and test their downstream consequences. In Experiment 1, we present evidence in support of distinct working memory representations for motor learning, and provide support for a transition from algorithmic to item-based-retrieval strategies in motor learning. In Experiments 2 and 3, we expose within-trial signatures underlying these distinct strategies. In Experiment 4 we characterize downstream consequences of different learning representations on behavioral flexibility and generalization.

## Results

**Experiment 1: dissociable strategies in motor adaptation.** Subjects performed a visuomotor rotation task (see Methods), where visual feedback was rotated relative to their reaching direction (Fig. 1a). Our hypothesis was that in low set size conditions (a small number of learning targets), subjects would primarily use a discrete RC strategy (i.e., a look-up table), while in high set size conditions they would use a parametric MR strategy (i.e., a rule-based algorithm). We used a $2 \times 2$ between-subjects factorial design, crossing Set Size and Rotation Magnitude. For Set Size, subjects were exposed to 2 or 12 possible target locations. In the 2T condition, one of two targets, separated by 180°, was pseudorandomly presented at each trial. In the 12T condition, target locations were pseudorandomly presented at 1 of 12 possible locations. For Rotation Magnitude, subjects experienced either a 25° or 75° rotation during the rotation block to test for signatures of MR. Feedback, in the form of a small visual cursor, was only provided at the end of the reach and was delayed by 2 s, a manipulation which limits implicit motor adaptation[28–31] to better isolate strategic learning.

Our first analysis focused on RT. We hypothesized that in the high Set Size group (12T), RT would be modulated by Rotation Magnitude throughout learning, consistent with the idea that subjects were parametrically mentally rotating their motor plans on each trial. In the low Set Size group (2T), we expected this effect only early in learning, while subjects discovered the "structure" of the task, and subsequently we expected RTs to converge in the two 2T groups due to the caching of responses. Our $2 \times 2$ design could test these specific predictions: MR in all four groups early in learning, and a shift to RC, for the 2T groups, late in learning.

Consistent with these predictions, Rotation Magnitude affected RT in all groups early in learning, but only the 12T groups late in learning (Fig. 2a). To quantify this, we examined the means of median RTs over 6 cycles between early learning (first 6 cycles)

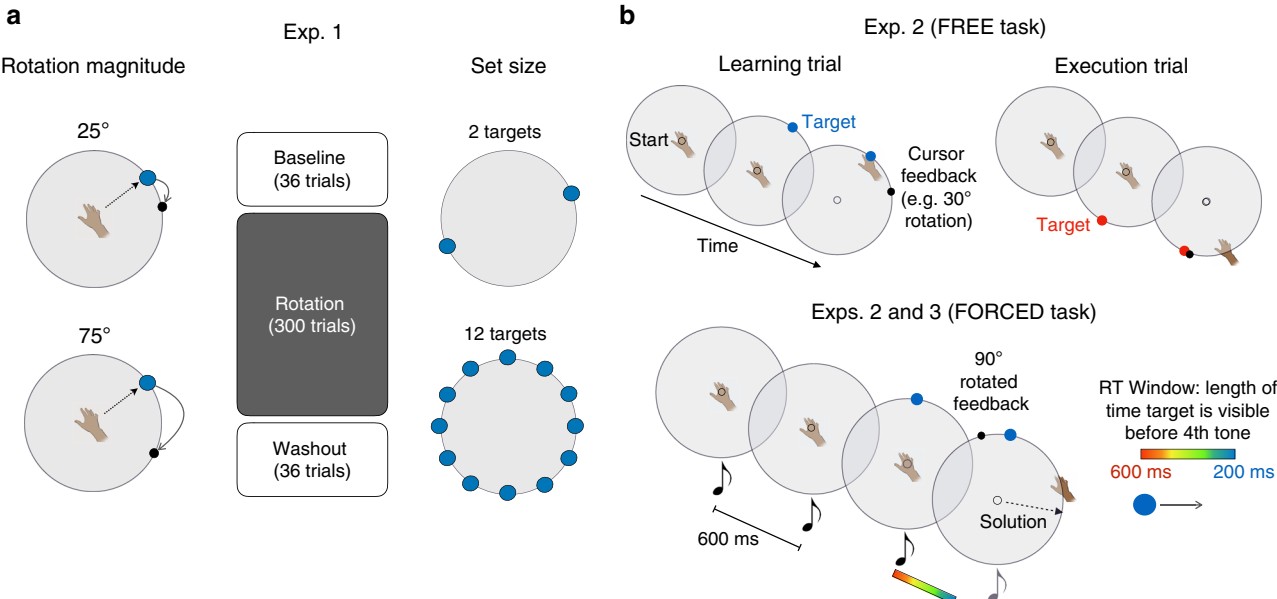

**Fig. 1** Experiment 1–3 task design. **a** Experiment 1: subjects performed a standard visuomotor rotation task, attempting to land a rotated visual cursor on a target. A between-subjects, 2 × 2 design was used, crossing the factors Rotation Magnitude (the size of the rotation in the rotation block; left) and Set Size (the number of possible target locations in the task; right). **b** Experiments 2–3: FREE task: subjects performed trial pairs consisting of learning trials (left) and execution trials (right). Thirteen rotation sizes were pseudorandomly presented (−90°:15°:90°). FORCED task: subjects were required to initiate their movement after target presentation and <100 ms after the fourth tone. Targets could appear in either 1 of 12 locations (Exp. 2) or 1 of 2 locations (Exp. 3). The time of target appearance was titrated to induce subjects to react with a distribution of RTs

and late learning (last 6 cycles). We defined a cycle as 2 trials in the 2T group and 12 trials in the 12T group to control for inherent difference between Set Size conditions in the number of exposures at each target location. RTs (Fig. 2c) were submitted to a three-way mixed factorial analysis of variance (ANOVA) with a within-subjects factor of Time (early vs. late learning), and between-subject factors of Set Size and Rotation Magnitude. We note that the lack of Set Size effects early in learning are an example of how an algorithmic strategy can seemingly bypass Hick's Law (i.e., a log-linear increase in RT over set sizes[32]).

We found a significant main effect of Time ($F_{(76)} = 67.97$, $p < 0.001$), reflecting decreased RT in all groups over training. We observed a significant effect of Rotation Magnitude, suggesting that subjects employed MR—larger rotations resulted in longer RTs ($F_{(76)} = 8.32$, $p = 0.005$). We did not find a main effect of Set Size ($F_{(76)} = 0.90$, $p = 0.34$), but, critically, found a significant two-way Time × Set Size interaction ($F_{(76)} = 20.87$, $p < 0.001$) and a three-way Time × Set Size × Rotation interaction ($F_{(76)} = 4.56$, $p = 0.036$). Post hoc $t$-tests (Bonferroni-corrected) revealed an effect of Rotation Magnitude on late RT in the 12T groups ($t_{(38)} = 2.45$; $p = 0.019$), but not in the 2T groups ($t_{(38)} = 0.15$; $p = 0.88$). These results are consistent with the hypothesis that all groups may have used MR to find the task solution early in learning, and then either maintained that strategy for the remainder of the task (12T), or transitioned to RC (2T).

RTs in the baseline period revealed no significant difference with respect to Set Size or Rotation Magnitude (ANOVA, $p = 0.63$ and $p = 0.97$, respectively). Likewise, in the washout block we found no effect of Rotation Magnitude on RT ($F_{(76)} = 2.40$; $p = 0.13$) and only a marginal effect of Set Size ($F_{(76)} = 3.35$; $p = 0.07$). Given this trend in the washout block (Fig. 2c), one speculation is that differences in RT at the end of training carried over, perhaps due to habitual rather than task-related factors[33].

Differences in RT could not be attributed to task performance between conditions: all groups displayed similar asymptotic performance, with only subtle differences in the rate of learning (Fig. 2b). We computed the average movement angular error (i.e., rotation minus movement angle) over the first 6 cycles and last 6 cycles and submitted these values to a two-way repeated measures ANOVA. We found a significant main effect of Time on error ($F_{(76)} = 74.70$, $p < 0.001$), reflecting learning, but no significant effects of Rotation ($F_{(76)} = 0.23$, $p = 0.64$) or Set Size ($F_{(76)} = 1.25$, $p = 0.27$), nor an interaction ($F_{(76)} = 1.34$, $p = 0.25$). We found no interaction between Rotation × Time ($F_{(76)} = 1.18$, $p = 0.28$). We observed a significant Time × Set Size interaction ($F_{(76)} = 6.28$, $p < 0.014$), reflecting slight learning advantages in the 2T groups. There were no significant differences in movement error during baseline with respect to Set Size or Rotation Magnitude (ANOVA, $p = 0.96$ and $p = 0.66$, respectively). Thus, neither Rotation Magnitude nor Set Size significantly affected task performance, consistent with previous results[34,35].

Delayed feedback was effective for limiting implicit adaptation, as mean aftereffects across the sample were only 2.71° (Fig. 2b). This suggests that the vast majority of learning was driven by cognitive strategies rather than implicit learning[5]. Although aftereffects were subtle, there were significant differences in movement angle in the washout block with respect to Set Size and Rotation Magnitude ($F_{(76)} = 16.10$, $p = 0.002$ and $F_{(76)} = 9.95$, $p < 0.001$, respectively), though there was no significant interaction ($F_{(76)} = 1.67$; $p = 0.20$). Corrected $t$-tests revealed that only the 2T 25° group showed statistically reliable aftereffects ($t_{(19)} = 3.49$; $p < 0.001$; all other $p$'s > 0.09). Parsimony suggests that this effect was not due to implicit adaptation, as in that case significant aftereffects should also be present in the other conditions[34]. The result is more consistent with "use-dependent" learning, which describes a bias toward repeated movement directions: use-dependent learning would be more robust in 2T

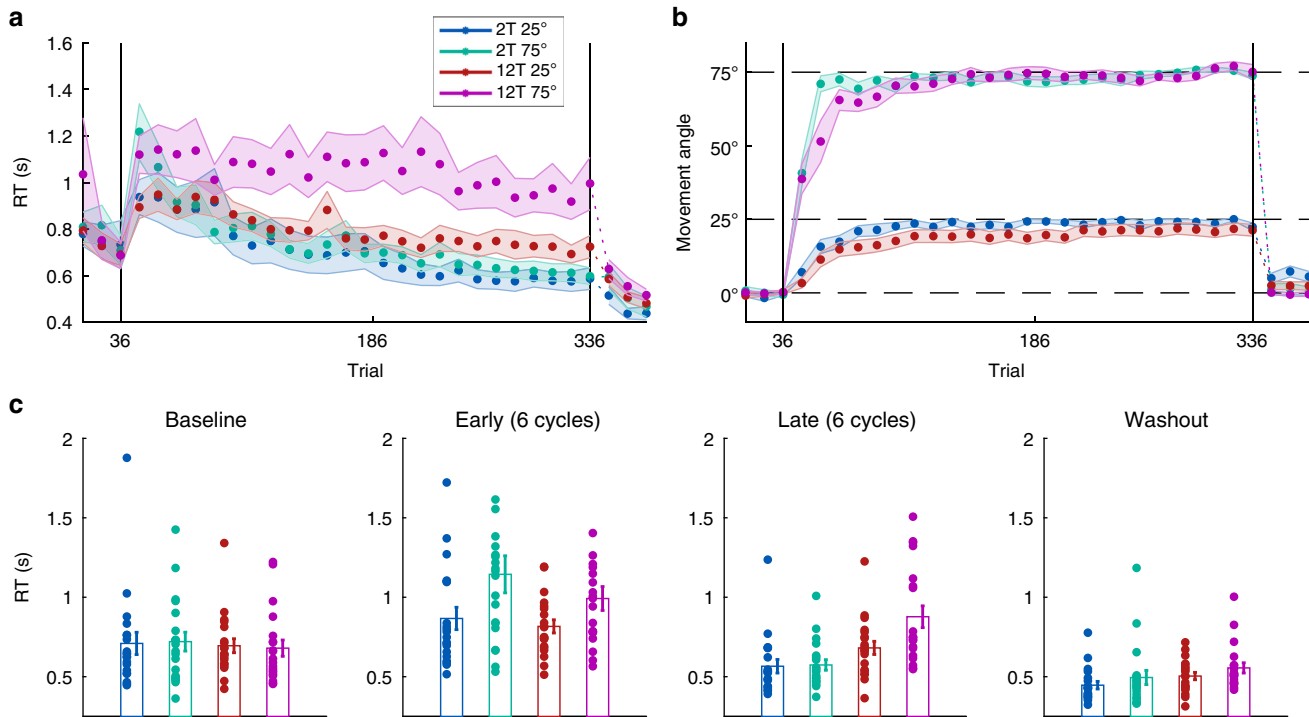

**Fig. 2** Experiment 1 results. **a** Time course of mean RTs, averaged over 2-trial and 12-trial cycles in the 2T and 12T groups ($n = 80$). Vertical lines delineate the task block (baseline, rotation, and washout). **b** Movement angle across the groups, averaged over 12-trial bins. **c** Mean of median RTs averaged over the cycles of trials in the beginning (early) and end (late) of learning, and the full baseline and washout blocks (see Methods). Error bars represent 1 s.e.m.

because only two responses are repeated, creating a stronger bias[36].

**Distributions of reach errors reveal strategy differences.** A secondary analysis (Fig. 3) also comports with dissociable discrete and parametric strategies. We analyzed subjects' sign errors (i.e., reaches in the wrong clockwise/counterclockwise direction relative to the correct response). Normatively, RC and MR strategies make distinct predictions about sign errors. In RC, sign errors should represent trials where participants accidentally retrieve the wrong response from memory (i.e., a working memory "swap" error[37]). This would constitute a −155° error in the 25° condition, and a −105° error in the 75° condition. In parametric MR, sign errors would most likely represent trials where participants accidentally flip the rotation angle, aiming with the correct magnitude relative to the target but the incorrect direction[13]. This would constitute a −25° error in the 25° condition, and a −75° error in the 75° condition.

Sign errors were designated as trials where subjects reached ≤−15° from the target location (i.e., opposite the correct positive response). Sign errors were relatively rare, and hence data were pooled. As predicted, swap errors were the more common error in the 2T groups, and flip errors were the more common error in the 12T groups (Fig. 3).

**Experiment 2: constraining RT reveals parametric strategies.** In Experiments 2 and 3, we wanted to confirm that the dissociable strategies characterized in Experiment 1 were valid models of within-trial cognitive computations. We adopted a "forced reaction time" task[38] which constrains the amount of time subjects have to prepare their responses. This procedure has recently been used to show how different processes proceed during motor learning[6,26]. Here, however, we wanted to rapidly induce movements to decode cognitive processes within a trial. Our hypotheses were as follows: interrupting MR should induce intermediate

movements representing partially rotated movement plans (Experiment 2), and interrupting RC should induce bimodal movement distributions with modes at each cached movement direction (Experiment 3).

We used a within-subject design with two tasks (FREE and FORCED tasks; Fig. 1b). In the FREE task, which was designed to capture the RT signatures of MR, subjects performed a series of trial pairs (Fig. 1b, top): in the first trial of each pair, the "learning" trial, subjects were instructed to reach directly at the displayed target and observe where the cursor landed. In the second trial, the "execution" trial, subjects were told to apply what they learned about the relationship between their movement and the resulting feedback and attempt to make the cursor terminate within the target (see Methods). Rotations ranged from −90° to 90° by 15° intervals and were pseudorandomly presented.

Subjects were generally accurate on execution trials, with reach angles tracking the magnitude of the rotation imposed on the corresponding learning trial (Fig. 4a; t-test on regression slopes: $t(31) = 30.57$, $p < 0.001$). As predicted, RT on execution trials linearly increased with the angle of movement relative to the target (Fig. 4b; t-test on regression slopes: $t(31) = 6.59$, $p < 0.001$), consistent with MR.

The FORCED task was designed to interrupt putative mental rotation (Fig. 1b, bottom), with the prediction that movement angle would be a linear function of RT. On each trial a countdown of four tones was played, and subjects were instructed to synchronize the initiation of their reach with the fourth tone (see Methods). Targets could appear in 1 of 12 locations. The moment of target appearance was titrated such that subjects had varying amounts of time with which to compute the target location, plan, and execute their movements. For rotation trials, a fixed rotation of 90° was imposed on the cursor, and subjects were thoroughly educated about the rotation beforehand.

We analyzed trials where subjects reached on time in accordance with the fourth tone ($\mu = 78.90\%$ of trials). RTs were

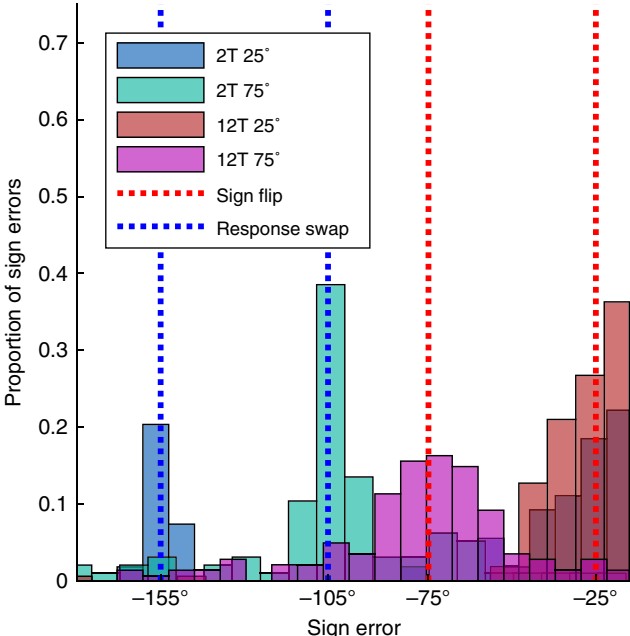

**Fig. 3** Sign error analysis. Histograms of pooled "sign error" data in each group. Vertical dotted lines represent predicted locations of "swap" errors (generating the wrong response given two choices) and "flip" errors (reaching relative to the correct magnitude of rotation but the wrong sign)

binned by 25 ms, from 0 ms through 400 ms, with the final bin including all RTs above 400 ms. We first identified the RT at which movement angles were reliably sensitive to the target location (at very short RTs, movements should be directed randomly since there is insufficient time to process the target[39]). Circular variance first significantly decreased ($t(31) = 2.28$, $p = 0.02$; Supplementary Fig. 1) from the 7th to the 8th RT bin (150–175 ms), suggesting that at RTs over 150 ms, subjects began to make non-random movements. This result is consistent with previous work[39].

After this time point, reaching angles monotonically increased with RT towards the solution (Fig. 4c; $t(31) = 14.17$, $p < 0.001$). This result echoes the rotation of a population vector in the motor cortex[24]. Moreover, the observed linear trend through intermediate movement directions (Fig. 4c, d) likely represents a behavioral correlate of intermediate states of mental rotation, confirming a fundamental assumption of analog computations[3,19].

**Mental rotation paces are correlated between tasks**. We assume that the FREE and FORCED tasks recruit the same parametric strategy. We derived a mental rotation pace parameter from each task (FREE and FORCED; see Methods). MR paces were strikingly similar between tasks ($t(31) = 0.19$, $p = 0.85$; Bayes factor = 7.81 in favor of the null). A significant correlation was found in subjects' mental rotation paces across tasks ($R^{Pearson} = 0.46$; $p = 0.008$; $R^{Spearman} = 0.41$; $p = 0.02$; Supplementary Fig. 2). This result suggests that the same computation is operative in our constrained and unconstrained RT contexts.

We note that FREE RTs (Fig. 4b) were well above those that produced equivalent movement directions in the FORCED task (Fig. 4c). We also note that the range of RTs observed in the FREE task is comparable to a similar study[21], and in the FREE task we found no correlation between the intercept of the regression, which reflects putative non-rotation RT, and its slope, which reflects MR ($R^{Pearson} = 0.03$; $p = 0.87$; $R^{Spearman} = 0.09$; $p = 0.64$). Thus, excess RT in the FREE task is likely the product of

processing unrelated to MR. Extra computation time could be the result of relatively low urgency in the FREE task.

**Control analyses confirm a parametric strategy**. We now address three alternative explanations for the observed rise in mean reach directions over RT in the FORCED task (Fig. 4c). First, because subjects made many random (i.e., uniformly distributed) movements at low RTs, and gradually made correct movements at higher RTs, a linear trend could appear as an averaging artifact. Critically, subjects' most frequent (mode) reach directions displayed intermediate values, gradually increasing with RT (Supplementary Fig. 3) and showing a significant linear trend ($t$-test on regression slope, $p = 0.002$), arguing against this particular averaging confound.

Second, non-random reaches could be limited to 0˚ (i.e., a prepotent response[40]) and 90˚, with the linear trend representing changes in relative frequency of each as RT increases. However, the mode analysis argues against this explanation; moreover, the full distribution of reach directions (Fig. 5a) shows no clear mode at 0˚. Subtle bimodality was indeed observed—subjects occasionally reached with an approximately correct magnitude of rotation but with an incorrect sign, echoing Experiment 1 (Fig. 3).

To further examine the distributions of reach angles over RT bins, we fit the data with two mixture models that accounted for random reaches (i.e., uniform distribution) and directed reaches (i.e., Von Mises distributions; see Methods). The first model (Free-$\mu$) allowed two mean parameters, one positive and one negative, to vary over RT bins (capturing correct responses and sign flips). The second model (Fixed-$\mu$) had fixed mean parameters at +90˚ and −90˚, consistent with a priori predictions of a discrete RC strategy (with swap errors).

The mean parameters of the Free-$\mu$ model showed evidence of MR (Fig. 5b): both the positive (purple) and negative (green) mean parameters gradually approached, respectively, +90˚ and −90˚ as RT increased. A regression on the fit mean parameters revealed a significant positive linear trend for the positive $\mu$ parameter ($p < 0.001$). Regression on the negative $\mu$ parameter revealed no significant trend ($p = 0.38$), though this result may be driven by the two deviant points (both >1.5 s.d. from the mean) in the first two bins. Figure 5c shows the full probability density functions of the Free-$\mu$ model. Critically, the Free-$\mu$ model provided a far superior fit vs. the Fixed-$\mu$ model ($\Delta$AIC = 1189, Supplementary Fig. 4), as the Free-$\mu$ model can better capture intermediate movements.

A third interpretation is that mental rotation could appear as the result of response substitution, where a response directed at the target location is gradually inhibited, while the rotated response gets excited[25]. Here, intermediate movements are explained by the averaging of simultaneously active motor plans[41]. Importantly, response substitution and MR make distinct predictions concerning movement speed: response substitution predicts a slowdown on trials where responses are averaged[42], whereas MR does not. Indeed, we found that movement time monotonically decreased ($\mu$ MT = 128.13 ms; Supplementary Fig. 5) while movement speed monotonically increased (Supplementary Fig. 6; $t(31) = 4.28$, $p < 0.001$) with increasing RT in the FORCED task, a result consistent with an MR strategy.

The different speed predictions of response substitution vs. MR can be captured in the length of a neural population vector, which is correlated with movement speed[43]. We conducted a computational modeling analysis using two neural population coding models. These models confirmed the different movement speed predictions, and provide further evidence that the speed data are compatible with MR but not response substitution (Supplementary Fig. 6). Furthermore, we note here that recent research has

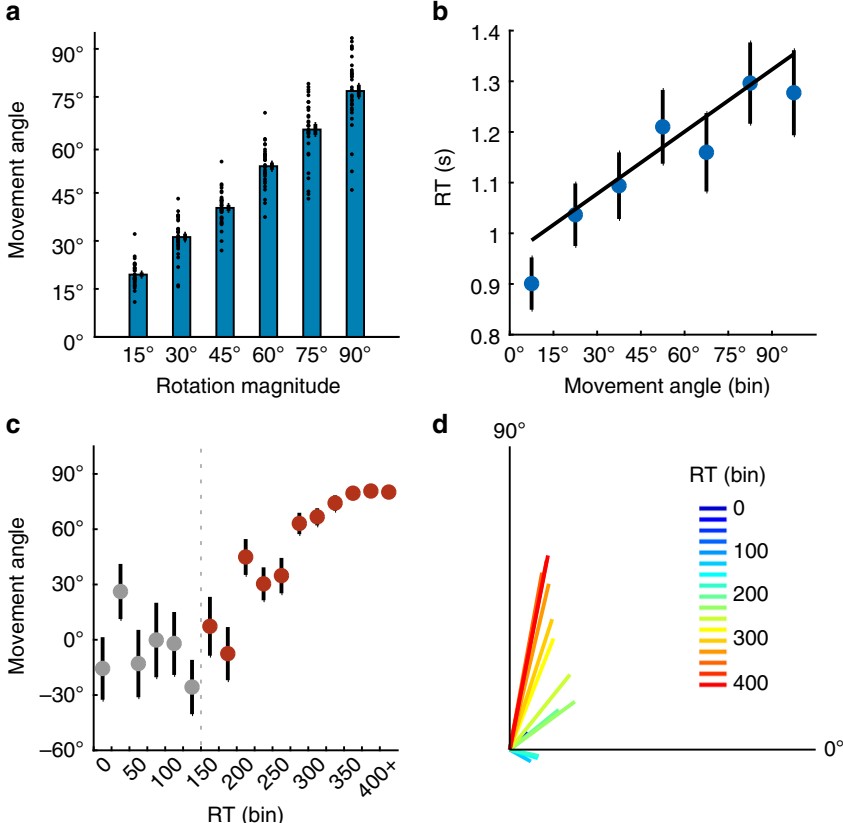

**Fig. 4** Experiment 2 results. **a** Absolute movement angles as a function of absolute rotation magnitude in the FREE task ($n = 32$). **b** Mean of median RTs as a function of movement angle (binned by 15˚). **c** Circular mean of movement angles as a function of RT (binned in 25 ms bins) in the FORCED task ($n = 32$). Gray dots represent RT bins where movements were deemed as primarily random (see Methods). **d** Pooled vector representation of data in **c**, where colors represent the associated RT bin (in ms). Error bars represent 1 s.e.m.

questioned the premise of involuntary averaging of parallel motor plans (see Discussion[44]).

**Experiment 3: constraining RT reveals a discrete strategy.** Experiment 3 was designed to provide further evidence of a discrete RC strategy. We used the identical forced-RT task as Experiment 2, but with 2 target locations instead of 12. We hypothesized that in this context, movements would follow a bimodal distribution reflecting cached responses.

Subjects could fully counteract the rotation with very short RTs: circular means of the movement angles (Fig. 6a) reveal an abrupt jump from highly variable movements at short RTs (<225 ms) to consistent movements at the rotation solution (90°). The vector plot (Fig. 6b) does not show the several highly variable intermediate values seen in the circular means, suggesting that those are likely the result of circular averaging of extreme values (e.g., −90˚ and +90˚; see model fitting below). The distribution of reaching directions (Fig. 7a) is consistent with a bimodal response distribution: subjects' responses were concentrated near the solution (+90˚) and its opposite (−90˚).

Unlike Experiment 2, we did not find a linear trend of movement angles across RT bins (t-test on regression slope, $p = 0.20$; Supplementary Fig. 3). We also fit the Free-$\mu$ and Fixed-$\mu$ mixture models to these data. Critically, the Von Mises mean parameters for the Free-$\mu$ model appeared to saturate immediately, suggesting no mental rotation (Fig. 7b, c). In fact, linear regression on the positive mean parameter revealed a slight negative slope ($p = 0.004$), which is the wrong direction for mental rotation, and regression on the negative mean parameter revealed no trend ($p = 0.36$). In contrast to Experiment 2, the

Fixed-$\mu$ model provided a far superior fit to the data ($\Delta$AIC = 1941, Supplementary Fig. 4). This comports with RC since there is no need for free mean parameters, and thus the model is not penalized for unnecessary complexity.

We also directly compared the 12-target and 2-target forced-RT conditions. First, we performed serial comparisons between circular mean reaching directions at each RT bin after the 150 ms bin. Means were significantly different in the 10th–13th RT bins (independent $t$-tests, all $p$'s < 0.05), spanning 200 and 300 ms. In the 12-target condition, all bins after and including the 7th bin were significantly different from 90˚ (one-sample $t$-tests, all $p$'s < 0.01). In the 2-target condition, movement angles were not significantly different from 90˚ starting at the 8th bin onward (one-sample $t$-tests, all $p$'s > 0.05).

**Mental rotation pace in FREE task predicts learning RTs.** By intermixing different rotation sizes, the FREE task in Experiment 2 provided an estimate of subjects' mental rotation paces (Fig. 4b). If our hypothesis is correct, the average mental rotation pace from the FREE task should correspond to observed RT differences between the 75˚ and 25˚ conditions from Experiment 1, and this correspondence should hold for 12T conditions in early and late learning, but only early in 2T conditions. The regression line predicted by the FREE task was consistent with MR occurring in all subjects early in learning (Fig. 8a; slope difference between FREE task distribution of slopes and early learning regression slope for 12T groups: $t(31) = 1.37$, $p = 0.18$; 2T groups: $t(31) = 1.55$, $p = 0.13$), but only the 12T groups late in learning (Fig. 8b; FREE task slope difference from 12T groups: $t(31) = 0.78$, $p = 0.44$; 2T groups: $t(31) = 6.16$, $p < 0.001$). This

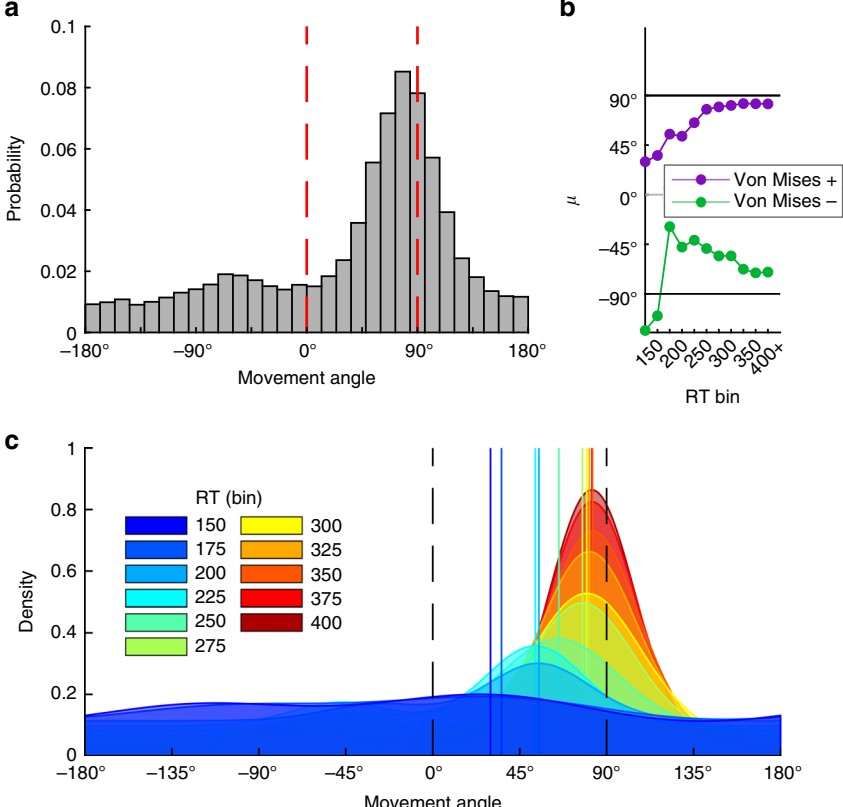

**Fig. 5** FORCED task full data distribution and model fits. **a** Full distribution of movement angles across all RTs in the FORCED task. **b** Fitted mean parameters from the "Free-$\mu$" mixture model (starting at the 150 ms bin) for directed reaches with a correct sign (Von Mises+, purple) and a flipped sign (Von Mises−, green). See Methods for model details. **c** PDFs of model fits in each RT bin (in ms). Vertical lines represent the fitted mean parameter (Von Mises+) for each corresponding RT bin

result is consistent with a set size-dependent shift from a parametric algorithm to a discrete retrieval strategy (Fig. 8c[15,27]).

**Experiment 4: generalization is affected by type of strategy.** How do different working memory representations for visuomotor learning affect generalization when conditions change? In Experiment 4, we investigated how different strategies would affect learning transfer. We reasoned that subjects using RC would show diminished generalization relative to subjects using MR. This could occur because under an RC regime, specific local instances are learned, whereas under an MR regime, a global rule (or structure) is learned and can be applied indiscriminately.

Subjects were trained on a 45° rotation in a constrained region of the workspace, with either 2 targets (2T) or 8 targets (8T), and the width between the furthest targets matched between conditions (Fig. 9; see Methods). After a brief rotation training block, subjects experienced a generalization test that tested transfer to novel targets. While this experiment could not directly infer subjects' learning strategies as in Experiments 1–3, we reasoned that the set size manipulation would bias subjects toward either RC (2T) or MR (8T).

As predicted, subjects in the 2T group showed more narrow generalization vs. the 8T group (Fig. 10a, b). We performed a trial-by-trial regression analysis on subjects' movement angles toward the generalization targets (Fig. 10c; Supplementary Fig. 7; see Methods). We found that the amount of practice (i.e., a trial number regressor) predicted an increase in movement angles (toward the correct response) at the generalization targets for both the 2T ($t(14) = 2.73$, $p = 0.02$) and 8T groups ($t(16) = 2.35$, $p = 0.03$), suggesting that generalization increased with time.

Consistent with our main hypothesis, the distance of generalization targets from the nearest training target negatively impacted movement angles in the 2T group ($t(14) = 3.06$, $p = 0.009$) but not in the 8T group ($t(16) = 0.10$, $p = 0.92$), and regression coefficients between groups were significantly different ($t(30) = 2.41$, $p = 0.02$). Thus, the cognitive strategy recruited for visuomotor learning shapes how the newly learned behavioral policy is generalized.

## Discussion

The role of working memory in motor learning is not well understood, though it is clear that controlled, deliberative processes are important[5,7,45]. Here we characterize two cognitive strategies in a visuomotor rotation task—discrete RC and parametric MR. MR manifests as a linear relationship between rotation magnitude and RT, consistent with classic mental rotation[3]. In contrast, RC manifests as a look-up table of S-R (stimulus-response) relationships[14,16], consistent with capacity-limited working memory[18,46].

How do parametric vs. discrete working memory representations relate to long-term skill acquisition? One useful analogy here could be the dissociation of model-based and model-free reinforcement learning[47], where the former relies on an explicit model of transition probabilities between responses and sensory states, and the latter merely reinforces rewarded actions. One speculation could be that model-based computations, which could be analogous to parametric motor learning strategies, are themselves made automatic over time[48]. This would suggest that the transition from parametric to discrete strategies we observed (Fig. 8) could represent mental rotation itself becoming

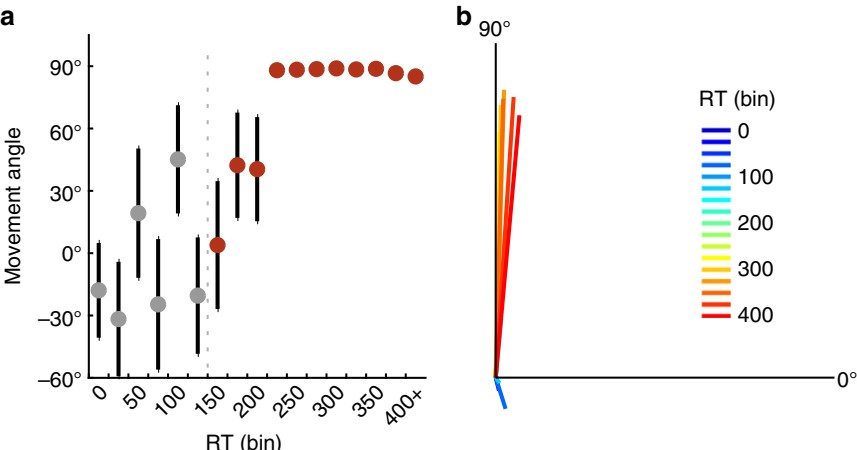

**Fig. 6** Experiment 3 results. **a** Circular mean of movement angles as a function of RT (binned in 25 ms bins) in the 2-target FORCED task (*n* = 10). **b** Pooled vector representation of data in **a**, where colors represent the associated RT bin (in ms). Error bars represent 1 s.e.m.

automatic. However, if a complex computation like mental rotation becomes automatic, it is hard to discern if the computation can still be said to be operative—it could be that the responses reflecting the computation have been cached, and the computation is thus no longer needed[27,49]. This kind of caching could represent an intermediate form of processing that lies somewhere between model-based planning and model-free reinforcement learning. In this framework, difficult computations need not be discarded during skill learning, but rather come to be bypassed during response preparation. Studies using extended training could further test this hypothesis[26].

Several studies on long-term visual mental rotation (using the task originally described in ref. [3]) find that although practicing mental rotation for weeks leads to an exponential reduction in overall RT, the mental rotation effect (i.e., RT as a linear function of rotation magnitude) diminishes only subtly[50–52]. Provost et al.[15] showed that repeatedly performing mental rotation on a small number of objects leads to the disappearance of mental rotation effects (perhaps consistent with RC), but when a large number of items are used during training, RT effects persist. These results in the visual domain overlap with our results, suggesting homologous mechanisms of spatial cognition. Moreover, these effects offer strong empirical support for Logan's theory of automaticity[27], whereby skill learning involves a transition from algorithmic to retrieval-based strategies.

We also note that the results of our 12-target forced-RT task (Fig. 4c, d) provide novel support to a critical prediction of analog cognitive computations, namely that mental rotation proceeds through intermediate states[3]. Intermediate states have been implied from RT measures[19] or neural recordings[24], but have not been rendered in overt behavior. Our results suggest that a mental rotation-like operation can drive volitional re-planning of a movement goal via a continuous sweep through direction space at a constrained pace. We note that this mechanism does not require that neurons necessarily be directionally tuned, but could instead represent a low-dimensional projection of a high-dimensional representation onto the two-dimensional angular space prescribed by our task[53].

Results in motor cortex[24] were challenged by an alternative explanation, which posits that the gradual "averaging" of an initial motor plan at 0° with a second plan at 90° could give the appearance of mental rotation[25]. This response substitution account has been supported by behavioral findings where subjects appear to average co-active motor plans in "go-before-you-know" paradigms[54] or "target-jump" saccade tasks[42], where goals are cued late in a trial. This averaging hypothesis has been challenged

by recent findings suggesting that nominally averaged plans may be the product of a decision-making process optimizing a single movement plan to account for multiple goals[44,55]. In a recent study[44], Wong and Haith[44] used a go-before-you-know task where subjects had to initiate a movement between two targets before one was cued as the goal. Critically, when moving relatively slowly (average movement times ~600 ms), subjects often reached at an angle between two competing targets until the goal target was cued, after which they swerved to the correct choice. However, when moving quickly (average movement times ~300 ms), subjects often moved in a straight line to one of the two options, making no corrections after the cue. Critically, not only did our FORCED task have a single goal (a 90° reach) instead of multiple goals, but, more importantly, subjects elicited rapid reaching movements (average movement times ~130 ms) well below the threshold for putative plan optimization[44]. Taken together, our task design and results suggest that neither obligatory averaging of co-active motor plans[54] nor strategic plan averaging[55] describe our results, and we maintain that transformation of a motor plan is the most parsimonious explanation.

One trend in our data is the tendency for subjects to "underrotate" when they were putatively using a parametric MR strategy. This can be observed in Experiment 1 (12T groups; Fig. 2b), both the FREE and FORCED tasks of Experiment 2 (Fig. 4), and the 8T group's training target behavior in Experiment 4 (Fig. 10a). In contrast, when performing discrete RC, subjects appear to reach all the way to the solution (2T groups Figs. 2b and 6). We do not have a clear result that speaks to this issue. One speculation is that due to the extra computation time needed for mental rotation, a latent urgency signal may drive subjects to initiate their movement before they finished rotating, perhaps reflecting a speed–accuracy trade-off. Another detail in our data is that MR in the 12-target FORCED task (Experiment 2) appears to start with a rapid jump followed by a slower rotation (Fig. 5b). This suggests that the MR process may involve a sequence of rotations with varying magnitudes.

In terms of the neural correlates of MR and RC, we present several hypotheses. For MR, parietal areas—the putative locus of mental rotation and similar sensorimotor transformations[56–59]—could feed a shifting movement goal to motor cortex, which would explain the observed intermediate movements when RT was cut short. Consistent with this hypothesis, Anguera et al.[12] found overlapping activation in inferior parietal and dorso-lateral prefrontal cortices during both visuomotor learning and mental rotation. Discrete RC, on the other hand, being a form of capacity-limited item-based working memory, could rely on

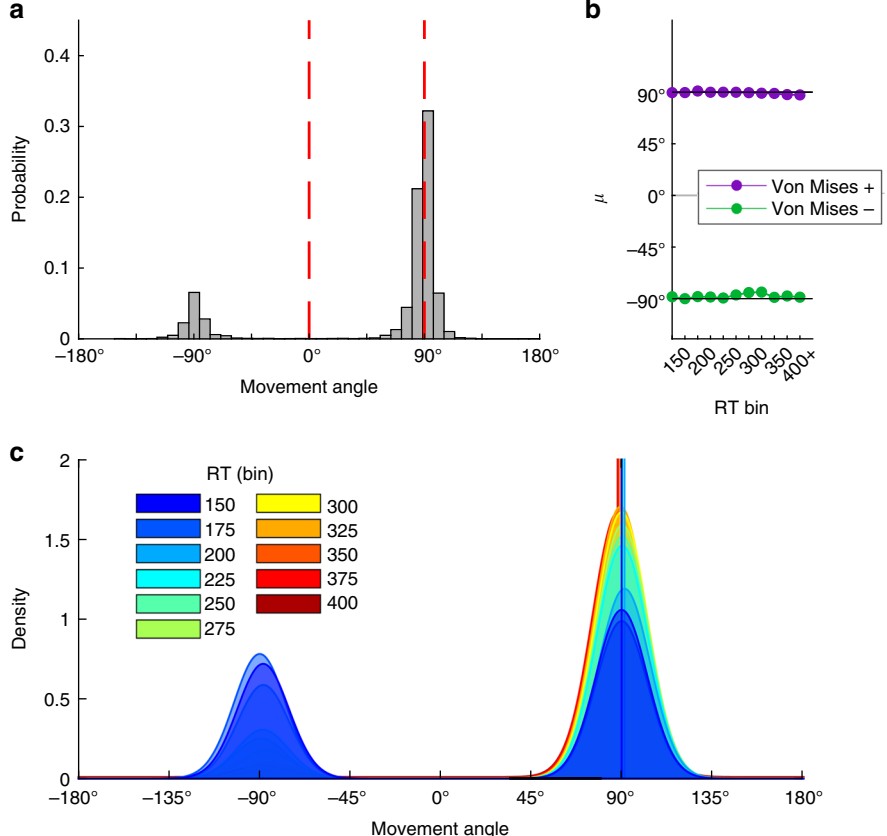

**Fig. 7** The 2-target FORCED task full data distribution and model fits. **a** Full distribution of movement angles across all RTs in the 2-target variant of the FORCED task. **b** Fitted mean parameters from the "Free-$\mu$" mixture model (starting at the 150 ms bin) for directed reaches with a correct sign (Von Mises +, purple) and a flipped sign (Von Mises−, green). See Methods for model details. **c** PDFs of model fits in each RT bin (in ms)

maintaining S-R relationships in dorso-lateral prefrontal cortex[17]. Future studies could test these hypotheses using brain imaging or stimulation.

A dissociation between parametric and discrete memory representations is a theme in other learning and memory domains. One example is human mathematical reasoning, where there are behavioral and neural dissociations between numerical computations learned by rote ("two times two is four") vs. flexible parametric operations performed on a mental number line represented in fronto-parietal spatial working memory regions[60,61]. Here we present an analogous dissociation, showing evidence for both the maintenance of discrete motor responses and the parametric manipulation of such responses. Action plans related to these strategies may be held and manipulated in working memory, perhaps in a manner consistent with visual, auditory, and tactile representations. Furthermore, the particular cognitive strategy that a subject adopts during learning appears to have downstream consequences for speed–accuracy trade-offs and behavioral flexibility.

## Methods

**Participants**. A total of 158 right-handed subjects (age range 18–34, 73 women) were recruited from the research participation pool maintained by Princeton University in exchange for course credit or monetary compensation. Handedness was verified using the Edinburgh handedness inventory[62]. All subjects participated in accordance with the university's institutional review board and provided written, informed consent. In Experiment 1 (n = 80), 20 subjects were used per condition. A single subject was excluded for disregarding the key task instruction of attempting to land the cursor on the target (asymptotic movement error >6 s.d. from the mean). We note that the sample size was not determined by a power analysis, although it is consistent with sample sizes in other studies using similar tasks[35,44,54]. In Experiment 2, a power analysis was used (alpha = 0.95) and revealed that a sample of 19 subjects would replicate the effect size of a relevant

correlation result (d = 0.66; correlation of mental rotation paces in ref. [22]). To be conservative, we sought to approximately double it, and hence we recruited a sample size of 32 (given counterbalancing). This is also in the range of a previous study that used a similar within-subject design and a somewhat similar task[22], which had a sample size of 26. A power analysis was used to determine the sample size in Experiment 3, which aimed to test the regression effect seen in Experiment 2 (d = 2.60; regression on movement angle × RT). The necessary sample size was 5, but, again to be conservative and counterbalance rotation signs, we recruited a sample of 10. For Experiment 4, we did not have a salient comparable analysis with which to conduct an a priori power analysis (i.e., regression weights on generalization probe distance vs. movement). However, in a previous study where we modeled generalization[35], we used a sample of 15 subjects. Thus, we opted to use 36 subjects in total (18 per group), a number which allowed for symmetric counterbalancing according to the task design.

**General experimental procedures and analysis**. In all experiments, subjects made rapid, center-out, open-loop reaching movements to visual targets (5.0 mm radius) using a digitizing tablet, holding on to a digital pen with their hand in a power grip position and sliding the pen across the tablet (Intuous Pro; Wacom). The task was controlled by custom software written in MATLAB (Mathworks, Natick, MA; Psychophysics Toolbox). Hand position was sampled at 140 Hz. Stimuli were shown on a 21.5-inch LCD computer monitor (Planar), mounted horizontally 25 cm above the tablet, occluding vision of the hand. A small cursor (2.5 mm radius) provided endpoint feedback after each reach terminated. Analyses were conducted in MATLAB and R (GNU).

**Experiment 1**. Subjects (N = 80) made rapid, 7 cm movements to targets in a blocked reaching task using a 2 × 2 between-subjects design, crossing the factors rotation magnitude and number of targets ("set size"; Fig. 1a). For the rotation, two magnitudes were used, ±25˚ and ±75˚, with the sign of the rotation counterbalanced within each condition. For set size, subjects were exposed to either two targets (2T) or 12 targets (12T). In the 12T condition, target locations were pseudorandomly presented at 12 possible locations (0˚, 30˚, 60˚, 90˚, 120˚, 150˚, 180˚, 210˚, 240˚, 270˚, 300˚, 330˚), where the same location was never repeated in consecutive trials. In the 2T condition, two targets were presented, and the locations were randomized across subjects to include one position from the 12 possible locations above and its opposite 180˚ away (Fig. 1a). Within a subject, targets were

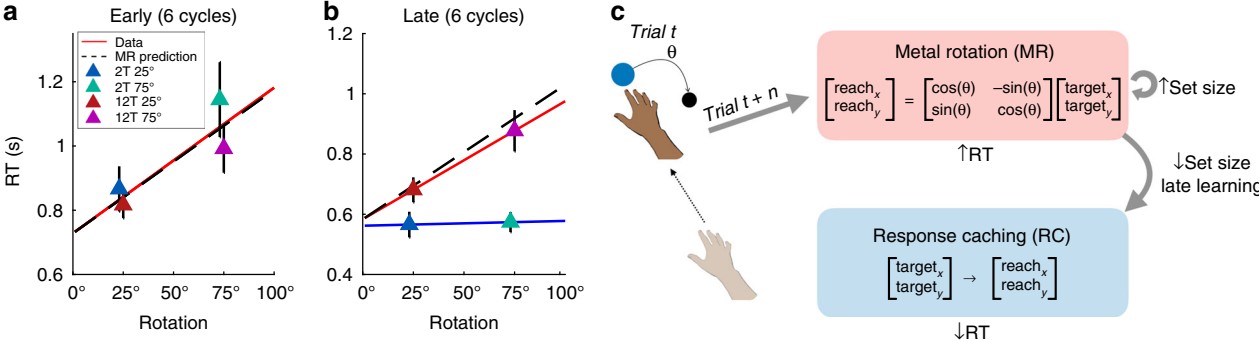

**Fig. 8** Mental rotation pace from FREE task matches RT data in Experiment 1. **a** Mean of median RTs in early learning from Experiment 1 (triangle markers; $n = 80$) with a regression line, for visualization, on all four groups' data (red), and the regression line predicted by the mental rotation pace gleaned from the FREE task ($n = 32$) using the same intercept (dashed black). **b** Same as **a** on late learning data, with separate regression lines for the 12T groups (red) and 2T groups (blue). **c** Schematic describing a parametric to discrete learning shift, which is mediated by set size. Error bars represent 1 s.e.m.

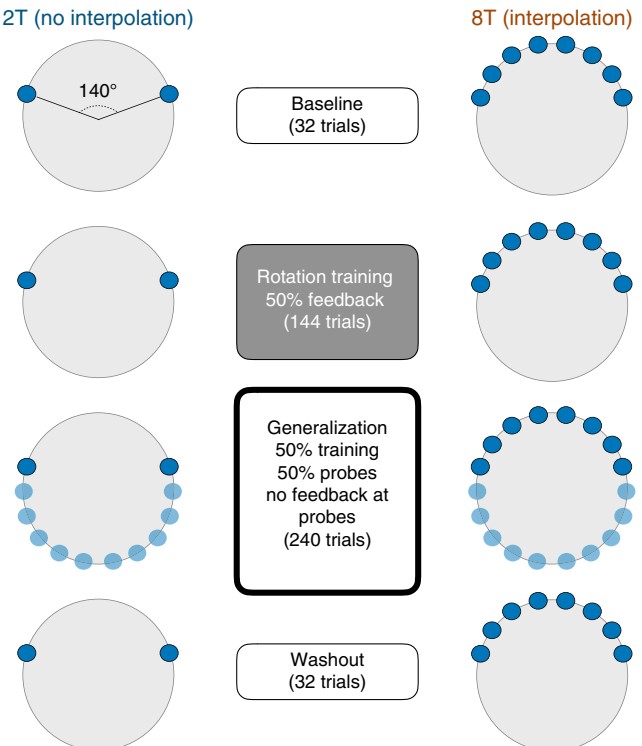

**Fig. 9** Experiment 4 design. Subjects learned to counter a 45° rotation in either a 2T learning condition (left) or an 8T learning condition (right). In a generalization block, novel targets were presented

presented pseudorandomly, where the same target was not repeated more than twice in a row. At the start of every trial, to position their hand in a central "start" position (5 mm radius), subjects used a dynamic visual ring that was scaled based on their distance from the start. After holding the start for 500 ms, a visual target appeared and cursor feedback was removed.

In a baseline block (36 trials), subjects reached to the targets with veridical endpoint feedback. At the start of the subsequent rotation block, the rotation was abruptly applied and was maintained for 300 trials, all with endpoint feedback. Finally, in the washout block, subjects were told to cease any strategy they may have adopted to counter the rotation and reach directly for each target. At the start of the experiment, subjects were told to do their best to land the cursor on the target. If the cursor hit the target, a pleasant chime was sounded, otherwise an aversive buzzer was sounded. In the washout block a neutral sound was played after each reach. Critically, to limit implicit sensorimotor adaptation and better isolate strategic explicit learning, all feedback was delayed by 2 s on every trial[28–31].

Movement angles were computed as the angle of the hand, relative to the target, when it crossed the invisible target ring of 7 cm. All movement angles were rotated to a common reference frame about the 0° axis of the unit circle. In all experiments, RT was measured as the time elapsed from target appearance to the hand leaving the start circle (i.e., 5 mm from the center of the tablet), and MT was measured as the duration of the movement from the end of the RT to the time at which the hand crossed the invisible workspace ring. Lastly, trials where RTs exceeded three standard deviations above or below a subject's mean RT (1.30% of trials), or trials where movement angles exceeded three standard deviations above or below a subject's mean movement angle (0.87% of trials), were excluded from analyses (the latter exclusions were not performed for the sign error analysis in Fig. 3).

**Experiment 2.** Subjects ($N = 32$) performed two tasks (FREE and FORCED; Fig. 1b), with the order of completion counterbalanced, as well as the rotation sign used in the FORCED task (−90° vs. 90°). The FREE task was designed to verify the classic signature of mental rotation, and the FORCED task was designed to interrupt mental rotation.

**FREE task.** At the start of every trial, subjects used continuous cursor feedback to position their hand in a central "start" position (5 mm radius). After holding for 400 ms, a visual target appeared and cursor feedback was removed. Participants were instructed to make a rapid, straight shooting movement to the target. The trial concluded when the hand crossed an invisible ring (8 cm radius), at which point feedback of the cursor was provided at the location where the hand crossed the ring. Slow MTs were discouraged: If MT exceeded a 700 ms limit, a "too slow" warning was delivered visually to the subject by the task software. In line with the instructions, subjects moved straight and rapidly, with a mean MT of 208.32 ± 8.54 ms (standard error of the mean).

Subjects were instructed that they would be performing a number of trial "pairs" (Fig. 1b, top): in the first trial of each pair, the "learning" trial, subjects were instructed to reach directly at the displayed target and observe where the feedback cursor landed. In learning trials, the target was blue and appeared in one of four off-cardinal locations (10°, 100°, 190°, 280°). In the second trial of the pair, the "execution" trial, subjects were told to apply what they learned about the relationship between their movement and the resultant feedback and attempt to make the cursor terminate within the target. In execution trials, the target was red and appeared in one of the three locations in which the learning target did not appear. Target locations were pseudo-randomized within and between trial pairs. Subjects performed 140 trial pairs. This task was modeled after a previous study[22], but with the distinction that subjects were not provided with an explicit symbolic cue regarding the exact solution to the rotation; instead, subjects had to determine the rotation's size and sign themselves, so that our task echoed canonical visuomotor adaptation paradigms[4].

Rotations used in the learning trials ranged from −90° to 90° by 15° intervals. Rotations were pseudo-randomized throughout the task, and each rotation was seen on 10 different trial pairs, except for the 0° null rotation, which was seen on 20 trial pairs. Each subject received an individualized schedule of rotation magnitudes and target locations.

Analyses were conducted on execution trials only, limited to trials where there was an imposed rotation. Movement angles were computed as the angle of the hand, relative to the target, when it crossed the invisible target ring. All movement angles were rotated to the same 0° target axis and matched to a single sign for analysis (see Experiment 1). The first analysis was a regression on subjects' reaching angles, relative to the imposed rotation, on execution trials. We chose to use the absolute reaching angle since on a significant number of trials (8.71%), subjects approximated the magnitude of the rotation but misinterpreted the sign (i.e., were within 15° of the "flipped" solution). No significant difference in mean RTs was found between correct and "flipped" execution trials ($t(31) = -0.99$, $p = 0.33$), suggesting that subjects did not hesitate and change their mind on flipped trials, but simply misremembered the proper clockwise/counterclockwise direction

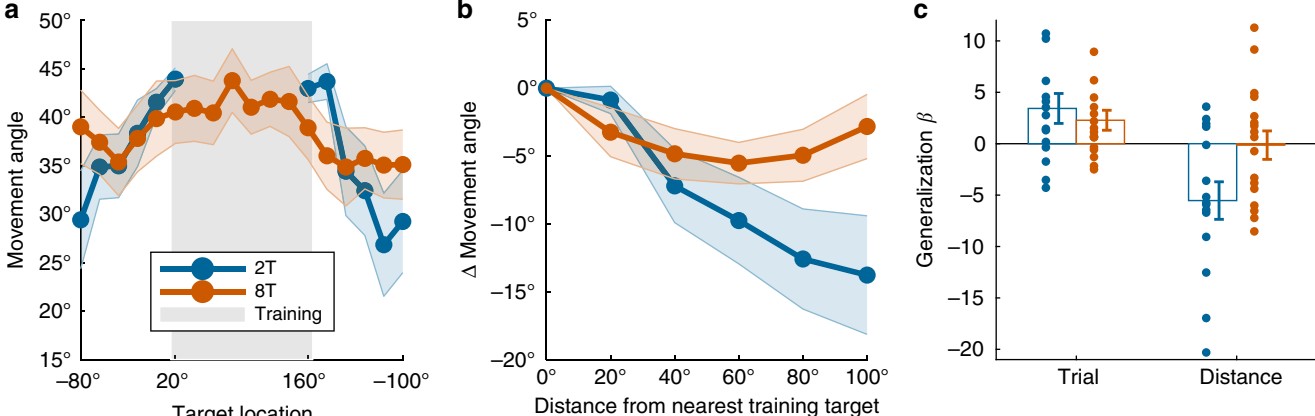

**Fig. 10** Experiment 4 results. **a** Mean movement angles in degrees ($n = 32$) in the generalization block targets and training block targets (gray shading). **b** Mean change in movement angle (degrees) relative to mean movement angles at the nearest training targets. **c** Results of regression analysis, showing effects of trial number and generalization target distance on movement angles in the generalization block. Shading and error bars represent 1 s.e.m.

at the onset of the execution trial (see Results and Discussion for a further discussion of sign errors).

A one-sample $t$-test was performed on the fitted slopes resulting from the regression of rotation magnitude and movement angle to test for the presence of a reliable trend. The second analysis was a regression of each subject's realized movement angles onto corresponding RTs. A one-sample $t$-test was performed on the fitted slope values to test for the presence of a reliable trend. Moreover, each subject's slope derived from this regression analysis served as their mental rotation "pace", in the units of milliseconds per degree[22]. We also performed a similar linear regression using the imposed rotation angles as the predictor variable to confirm that it echoed the regression using the actually realized movement angles; both analyses yielded comparable results, though the movement angle regression is likely to be a better estimate of a subject's true mental rotation pace[21].

**FORCED task**. The FORCED task utilized a modified forced-response-time paradigm to interrupt putative mental rotation (Fig. 1b, bottom;[38,39]). Like the FREE task, at the start of every trial subjects used continuous cursor feedback to position their hand in a central "start" position. Once subjects were positioned in the start, a countdown of four tones was emitted from the computer speakers (Logitech). The tones were played 600 ms apart, and subjects were instructed to try and synchronize the initiation of their reach with the fourth tone, effectively "replacing" that tone. The experiment and instructions were deliberately designed to emphasize early movements over late ones: if the subject initiated their movement >100 ms after the fourth tone, the screen was blanked and a "Too Late" message was displayed. Subjects could initiate their movement any time after target appearance, to encourage early RTs. However, if they moved before the target appeared the screen was blanked and a "Too Early" message was displayed. Similar versions of this forced-response-time task have been previously used to study the effect of restricted RTs on visuomotor learning and movement preparation[38,39].

Subjects were instructed to prioritize reaching on time, and, secondarily, to try and land the cursor on the displayed target. On each trial, the targets could appear in 1 of 12 off-cardinal locations (10°, 40°, 70°, 100°, 130°, 160°, 190°, 220°, 250°, 280°, 310°, 340°). Endpoint cursor feedback was presented after the subjects passed the invisible target ring, and the appearance of the feedback was delayed by 500 ms. Like Experiment 1, this brief delay was added to inhibit implicit adaptation to the imposed rotation and thus isolate the cognitive re-aiming process[28–31]. A briefer delay was used in this experiment because more trials were desired, and 500 ms delays have been shown to significantly attenuate implicit adaptation[28]. The delay manipulation was successful, yielding subtle, though significant aftereffects ($\mu = 1.98°$; $t(31) = 2.06$, $p = 0.048$).

The moment of target appearance was titrated such that subjects had varying amounts of time with which to compute the target location, plan, and execute their movements. These windows reflected the time elapsed between target appearance and the fixed moment of the fourth tone. Seventeen test RT windows were used, ranging from 200 to 600 ms by 25 ms intervals. An asymptotic RT window was also used, giving subjects up to 1200 ms to react. We reiterate that because subjects could move early in our variant of this task (i.e., any time after target appearance), the windows acted as "guides" rather than being perfectly predictive of subjects' realized RTs (i.e., subjects tended to move before they needed to; see Results). Finally, catch trials of 0 ms were used to ensure that subjects stayed on task and executed movements on time even if they had not perceived the target yet.

Subjects performed the FORCED task in three blocks. To get accustomed to the task, in the first block subjects performed 64 baseline trials with veridical endpoint cursor feedback, with pseudo-randomized RT windows and target locations. In the subsequent rotation block, subjects performed 624 trials, with 35 trials at each of the test RT windows, 14 trials at the asymptotic RT window, and 15 catch trials.

For the rotation block, a fixed rotation of 90° (or −90°, for counterbalancing) was imposed on the cursor. Subjects were thoroughly educated about the rotation before this block began. Subjects were told to try and counter the rotation and land the cursor on the target every trial. Due to the difficulty of the task, subjects were encouraged by a monetary bonus of up to $5 based on their performance in the rotation block, and were informed that missed trials (reaching too late or too slowly) would count against their performance score. Finally, in the aftereffect block, subjects performed 48 trials with a 1200 ms RT window, pseudo-randomized target locations, and no cursor feedback. Subjects were told to reach directly for the target on every trial of the aftereffect block. These data were later compared to the baseline data to get an estimate of any implicit adaptation.

All analyses were conducted on trials where subjects reached on time (78.90% of trials). Like Experiment 1 and the FREE task, movement angles were computed as the angle of the hand relative to the target when it crossed the invisible target ring. Importantly, subjects followed the instructions and made straight shooting movements: Movement times were rapid ($\mu = 128.13$ ms) and no feedback was provided during the reach, which helped to ensure movements were straight. Movement speed was computed by taking the average of the derivative of hand position from the time subjects left the start circle to the time they crossed the invisible target ring (due to the rapid "shooting" movement required by the task, average speed was used instead of peak speed to reduce noise in the estimate as subjects often reached peak speed after passing the target; we note that this particular approximation of speed did not influence the main results of the analyses related to movement speed).

Our first analysis involved investigating subjects' reach angles as a function of RT. RT bins were taken every 25 ms, from 0 ms through 400 ms, with the final bin including all RTs above 400 ms. Similar to previous results[39], in catch trials (0 ms RT window) subjects prioritized the timing demands of the task when no target appeared and moved relatively randomly around the circle, with some slight biases. A single "critical RT bin" was computed, after which reaches were determined to be primarily non-random (Supplementary Fig. 1). Linear regressions were performed on subjects' full distribution (i.e., no binning) of reach angles and RTs after the critical RT bin, and a $t$-test was performed on the resulting slopes to test for the presence of a significant trend. These slopes were used as the mental rotation "pace" parameters in further analyses.

Mental rotation paces were compared between the FREE and FORCED tasks in four ways. First, a one-sample $t$-test was performed to show that the null could not be rejected. Second, a Bayes factor was computed on the resulting one-sample $t$-value using the JZS (Jeffreys–Zellner–Siow) method[63] to quantify evidence for the null. Third, both parametric (Pearson) and non-parametric (Spearman) correlations were conducted between the values to test for a significant relationship that is robust to outliers. To confirm the robustness of this correlation, a secondary analysis that involved fitting a sigmoid to the data was used as an alternative method for extracting the pace parameter in the FORCED task (Supplementary Fig. 2). Lastly, two supplementary modeling analyses were conducted to test alternative interpretations, involving both a mixture model analysis (see below; Fig. 5) and a neural model which modeled movement speed and direction using a cosine-tuned population coding model (Supplementary Fig. 6).

**Experiment 3**. Subjects ($N = 10$) performed the FORCED task used in Experiment 2 (Fig. 1b), with one critical difference: only two target locations were used, where one was drawn from 1 of the 12 possible locations of Experiment 2 (FORCED), and the other target was its 180° counterpart. The particular pair of targets used, and the sign of the 90° rotation, were counterbalanced across subjects. All task instructions and analyses matched those described in Experiment 2. MTs were similarly rapid ($\mu = 111.25$ ms), and aftereffects were similarly small ($\mu = 3.17°$)

but did not reach significance ($t(9) = 2.12$, $p = 0.06$). For comparison purposes, the same "critical" RT bin was used in this experiment as that derived from Experiment 2 (RTs > 150 ms; see above).

**Experiment 4.** Subjects ($N = 36$) performed a reaching task (Fig. 9) that was identical to Experiment 1 in terms of basic trial design, visual stimuli, and feedback timing. In a baseline block (32 trials), subjects reached to visual targets with veridical feedback. In the rotation learning block (144 trials), subjects experienced a consistent 45° rotation (or −45°, for counterbalancing), and received cursor feedback on 50% of those trials. In the learning block, subjects were divided into two groups: a 2T group and an 8T group. In the 2T group, learning targets appeared at one of two locations 140° apart, with the specific pair of locations counterbalanced across subjects. In the 8T group, learning targets appeared at 1 of 8 possible locations, spaced equally in a 140° region, with the specific locations also counterbalanced across subjects.

The generalization block tested how subjects extrapolated their learning to a new region of space. On 50% of trials, targets appeared at one of the previously presented target locations, and in the other 50% of trials targets appeared at 1 of 10 equally spaced novel locations within the 220° region of the workspace lying outside of the 140° training region. To test generalization without allowing for new learning at the generalization targets, no feedback was given on generalization probe trials. However, feedback was shown on all trials where a previously seen learning target was presented. Thus, learning at the original training locations was maintained, and the overall probability of seeing feedback was matched between the generalization and learning blocks (the latter was done to limit the "context change" brought about by the generalization block). In the generalization block, subjects were instructed to reach to novel targets in a manner that would cause the cursor to hit the target if they had seen the feedback. In a final washout block, subjects were told to cease any strategy they were using to counter the rotation and reach directly for the presented targets.

Movement angles were computed in the same manner as Experiment 1. Given the limited number of feedback trials in the rotation learning block (72 trials), we first analyzed whether subjects successfully adopted a strategy to counter the rotation in this brief period and with the added interference of the no-feedback trials. We used an a priori learning criteria derived from a recent paper[64]: In each subject, a one-sample $t$-test was performed against 0° on movement angles over the last 4 trial cycles (optimal movement angle = 45°). Four out of 36 subjects showed non-significant asymptotic learning ($p > 0.05$), and were thus excluded from the generalization analysis. Following Experiment 1, trials where RTs exceeded three standard deviations above or below a subject's mean RT, or movement angles exceeded three standard deviations above or below a subject's mean movement angle, were excluded (1.15% of trials).

Generalization was analyzed as follows. First, subjects' movement angles were rotated to a common reference frame so that the learning target region lied between 20° and 160°. For visualization purposes, movement angle generalization functions were computed according to both the raw target angle (Fig. 10a) and the change in movement angle as a function of the target's absolute distance from the nearest learning target (Fig. 10b). For group comparisons, a trial-by-trial regression analysis was performed using movement angles on generalization trials (i.e., movements to novel targets) as the dependent variable, and four separate $z$-scored regressors: the trial number, the distance of the current target from the nearest learning target, the subject's RT, and the interaction of RT and distance. (Non-distance-related regressors were added to better isolate the main effect of target distance.) Two-sample $t$-tests were performed on the resulting beta values of interest (Fig. 10c).

For completeness, we also conducted a more traditional generalization analysis, where Gaussian functions are fit to each subject's generalization data[35]. However, because of nearly complete generalization in many subjects (especially in the 8T group), various combinations of the height, width, and offset free parameters can yield flat generalization functions, leading to unstable parameter estimation (see Supplementary Fig. 7). Thus, we opted for the more interpretable regression approach.

**Mixture model.** A mixture model was used to characterize data in the FORCED tasks from Experiments 2 and 3. In this analysis, we modeled reach data as a mixture of two circular normal (Von Mises) distributions representing both positive (correct) and negative (flipped) directed reaches, and a single uniform distribution that represented random reaches. The model had probability density function,

$$P = w_1 \frac{e^{\kappa\cos(x - \mu_1)}}{2\pi I_0(\kappa)} + w_2 \frac{e^{\kappa\cos(x - \mu_2)}}{2\pi I_0(\kappa)} + (1 - w_1 - w_2)U(-\pi, \pi), \qquad (1)$$

where $I_0$ is the modified Bessel function of order 0, $w$ is the weight given to each distribution ($w_1 + w_2 \leq 1$), $\kappa$ is the concentration parameter (which we fit as a single parameter between the two Von Mises pdfs), and $U(-\pi, \pi)$ is the uniform probability density function of the unit circle. The mean parameter in Eq. 1, $\mu$, represents the mean of a given Von Mises, including one mean for positively signed reaches and one for negatively signed reaches ($\mu_1 > 0$, $\mu_2 \leq 0$). Data were pooled

across subjects in each RT bin and parameters were optimized separately in each bin using maximum likelihood estimation, minimizing the negative log likelihood with the MATLAB function fmincon. Fifty randomized starting parameter values were used for each fit to avoid local minima, and Aikake information criterion values (AICs) were either summed over all RT bins after the critical 7th bin or compared separately at each bin for model comparisons. Two models were compared: in the Free-$\mu$ model, both $\mu$ parameters could vary freely. In the Fixed-$\mu$ model, $\mu$ parameters were fixed at our a priori prediction of $\mu_1 = +90°$ and $\mu_2 = -90°$. Thus, the Fixed-$\mu$ model had two fewer degrees of freedom than the Free-$\mu$ model.

**Code availability.** The code for the task and for data analyses are available from the corresponding author upon request.

## Data availability
The data that support the findings of this study are available from the corresponding author upon request.

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

## Acknowledgements

We thank Chandra Greenberg for help with data collection. We also thank Fulvio Domini, Justin Jungé, and Eugene Poh for helpful discussions and comments on the manuscript. This work was supported by the National Science Foundation (Graduate Research Fellowship to S.D.M.) and the National Institute of Neurological Disorders and Stroke (Grant R01 NS-084948 to J.A.T.).

## Author contributions

S.D.M. and J.A.T. designed the experiments. S.D.M. performed the research and analyzed the data. S.D.M. and J.A.T. interpreted the results and wrote the paper.

## Additional information

**Competing interests:** The authors declare no competing interests.

