## [Peer Review File · Nature Communications]

Reviewers' Comments:

Reviewer #1:

Remarks to the Author:

This paper examines different cognitive strategies used by human participants when learning a simple motor task (a visuomotor rotation). Experiment 1 shows that participants employ either a response caching, or mental rotation strategy to solve a visuomotor rotation task - with response caching favored when the overall number of responses that need to be remembered is small (a single rotation applied at only 2 possible targets), and mental rotation favored when many different responses may be required (a single rotation applied at many different targets). The authors go on to dissect the processes used in each condition, demonstrating that pressuring reaction time leads to very different patterns of failure under a 2T or 12T condition, further reinforcing the qualitative different strategies that the authors interpret are occurring in Experiment 1.

Overall, this is a highly thought-provoking and imaginative paper. The experiments are creative and elegant and the data and analyses are, on the whole, very clear. Most importantly, the results provide considerable clarity to an emerging understanding of how longstanding ideas in psychology are at play in popular motor learning paradigms.

One aspect of the paper that I remain slightly unconvinced of is the conceptual framing in terms of "working memory". The synthesis across psychology and motor learning is a considerable strength of the paper. However, it is still unclear to me in what sense both RC and MR are obviously instantiations of 'working memory'. In particular, RC is commonly conceived of in terms of 'model-free' learning. The authors appeal to Collins and Frank, who have argued that RL-like learning might reflect formation of working-memory-based representations, rather than necessarily corticostriatal learning mechanisms. but couldn't, in principle, the learning seen here be more corticostriatal-based than working-memory-based? The working memory narrative in the title feels a bit hasty/speculative...

The discussion ends with the statement (line 577) that "working memory may operate on internally generated movement plans much like it operates on any number of sensory and abstract representations". In what sense does working memory "act" on anything? Isn't it a representation that can be acted upon, by something that can also act upon movement plans? The connection between working memory and motor learning will be something many in the motor learning field have not thought about much and this paper has the potential to be very didactic in this regard, so being crystal clear about these kind of statements seems essential.

The 'interference effect' analysis (Figure 3B) is weak. The effect seems clear for the 2T groups. However, the comparison with the 12T group is inadequate. First, simply claiming a difference between groups based on demonstrating a significant effect for one group, but no significant effect for the other group, is flawed logic. There should instead be a group-by-delay interaction. However, the interaction comparison can't be done because there is no 1-back data point for the 12T group. That itself is a problem since there might be something privileged about 1-back trials that is driving the decline in compensation in the 2T groups and would also have enabled good performance in the 12T group, had there been any/enough 1-back trials. Also, if the method of compensation is simple response-caching, it's not clear that interference should necessarily manifest itself as a reversion towards 0 compensation. Inspecting distributions of errors might help clarify things, but overall I am skeptical that this analysis tells us anything about differing strategies of compensation in the two groups.

The Results contains quite a lot of methodological details. Given that a lot of quite different experiments are described, this makes for quite a lengthy Results section. The authors might consider

streamlining the results a bit by describing only the most crucial aspects of the experimental design and providing pointers to other sections for the more specific details (e.g. lines 277-289 reads like it belongs in the methods section).

Fig. 3A - "probability" doesn't quite seem the right label for the y-axis. I think it is really the probability of that error size given that an error occurred, not the overall probability of that error size. Perhaps "proportion of errors" would pose less risk of being misleading?

512 - hyphenation: 12-target forced-reaction-time task

Reviewer #2:

Remarks to the Author:

OVERVIEW

It is now well established that adaptation of reaching movements to a visuomotor rotation (i.e., rotation of a cursor representing the hand about the movement start point) involves two components: an explicit component, which can result in fast error reduction, and an implicit component, which is more gradual.

There is now a substantial amount of evidence that the explicit, or strategic, component involves a form of mental rotation whereby the participant reaches toward a location, or 'aim point', that is rotated away from the viewed target. Consistent with this interpretation, a number of previous studies have shown that this re-aiming strategy takes time to implement and that this time (referred to as reaction time or preparation time) appears to scale with the magnitude of the rotation that is implemented.

In the current paper, McDougale and Taylor describe three experiments that provide clear evidence that, at least under certain circumstances, there is another component of adaptation. In the first experiment, they show that when participants adapt to targets at only two locations (180° apart), as opposed to targets at a larger number of locations, they learn stimulus-response pairings for each target location. These responses can be implemented rapidly and do not involve mental rotation. In the second and third experiments, the authors used a "forced reaction time task" where performance is examined as a function of preparation time. They show that with a large number of target locations, where mental rotation is predicted to be involved, no rotation is implemented at very short reaction times and that the amount of rotation increases (up to approximately full rotation) as reaction time increases. In contrast, with only two target locations, where stimulus response pairs are predicted to be involved, participants select one of the two possible responses at very short reactions and select the correct response when reaction is larger.

This paper is very clearly written and easy to read. I have no concerns about the methods and results. All three experiments are very well designed, and all of the analyses are very well done. I especially liked the analysis of errors. For the introduction, I would have preferred a more direct consideration of the visuomotor task, leaving consideration of other tasks (i.e., the piano analogy which I was not fully convinced by) for the discussion. However, this is a matter of taste and I would not insist on any changes.

Although I really liked this paper, I have to admit that I did not find the results particularly surprising and it seems to me that the basic results are predicted by previous work (see below). Perhaps I am being unfairly harsh here and would welcome a rebuttal by the authors.

MAJOR COMMENTS

The idea that participants learn stimulus-response pairings when only presented with targets are two locations (separated by 180°) does not seem very surprising. Indeed, a similar effect has been shown for mental rotation, per se. Thus, Provost and colleagues (2013) have shown that, after extensive practice with a small set of stimuli, mental rotation becomes very fast and no longer depends on rotation angle. In contrast, with a larger set of stimuli, mental rotation becomes faster but continues to depend on rotation angle. Provost and colleagues concluded that "extensive practice with a small set of stimuli allowed participants to switch from a mental rotation strategy ... to direct retrieval from memory of the correct response associated with each stimulus". They also note that "the development of a retrieval strategy is generally consistent with Logan's (1988) instance theory of automaticity."

The authors cite the Provost et al. paper and do indicate that there is considerable overlap with their results. I guess I feel that the overlap is substantial that the current results are predictable from these previous results. (Note that the authors imply that the Provost et al. paper deals with "perceptual learning" but I am not convinced this is accurate.)

The current results also seem similar to a paper on biorxiv: Huberdeau, Krakauer and Haith, "Practice induces a qualitative change in the memory representation for visuomotor learning", <http://dx.doi.org/10.1101/226415>). Huberdeau and colleagues examined adaptation to a visuomotor rotation, manipulated preparation time, and used a small number of target locations (but did not contrast this with a large number of target locations as in the current study). Huberdeau et al. found that with extensive practice learning, or relearning, to reach under visuomotor rotations, the preparation time required to implement the strategic component (initially involving mental rotation) becomes shorter and shorter, to the point at which it can reasonably be referred to as 'automatic'. I am not sure of the status of papers on biorxiv. However, it seems to me that this paper should be discussed.

ADDITIONAL COMMENTS

My only minor comment concerns the use of the term "bootstrapping the learning curve", which I found very odd and not appropriate.

Reviewer #3:

Remarks to the Author:

Authors make a strong case in support of the involvement of two working memory strategies during the reaction to visuomotor perturbations; stimulus response caching and mental rotation. These results are interesting and novel for the case of visuomotor explicit adaptation. However, the involvement of stimulus response and algorithm driven responses in control and learning has been proposed before in the discussion about improvement in mental rotation (that could be explained by a transition to an automatic response (Provost et al., 2012, Tarr & Pinker, 1989), discussed by authors), with the mental line results that the authors present, and dealt with in the context of the instance theory of automaticity by Logan (1988). In fact, it is interesting to compare these results with the model of Logan, especially the timed response results that shed light on the underlying running processes. I assume that the algorithms involved in motor learning may vary as a function of task and therefore not sure whether this is a general model for motor learning or a specific model for visuomotor rotation.

specific points:

Page 6 115 – please indicate that this is a between-subject design

Page 7 – reaction time is likely to also change as a function of error and success

Page 7 - The Lack of consistency with Hick's law should be discussed

Page 8 - 12T 75deg during the late phase show increased reaction time and increased errors. lack of success may indicate increased cautiousness rather than a qualitative difference in strategy.

8 178 – are there reference in support of reduced implicit adaptation as a function of delay? It could be that the increased delay between response and feedback also affected the estimation of the after effects and led to an under-estimation of the effect.

Experiment 1 aims both to validate the overall idea of two strategies in explicit adaptation and to examine specific questions about the involvement of the mechanisms in certain cases (2t 75 and 12t 25). This logic seems a bit circular. A clear prediction about the two conditions in the middle may be helpful in this case.

9 202 the swap errors make sense, but the flip errors not. Is there any support for this prediction?

Can authors provide additional intuition for this prediction?

10 215 – can this effect explained by a time difference between the presentations (and therefore reflect a passive decay process)? Or alternatively by a switching effect between the targets?

12 – in exp 2 feedback was delayed by 500ms and not by 2s as before. This difference should be mentioned and discussed.

12 268 – study design assumes full generalization between perturbation target and subsequent aiming target. Could narrower generalization provide an alternative explanation to the results?

Review Response

We would like to thank the reviewers for their insightful and helpful comments. Their comments helped us clarify the narrative and emphasize why our study provides novel insights into understanding different mechanisms that may be at play in sensorimotor learning tasks. We have also included an additional experiment that explores how these different strategies have downstream consequences in the generalizability of learning.

The reviewer comments are in plain font, our response is in bold, and changes to the text are in quotations.

Reviewer #1 (Remarks to the Author):

Overall, this is a highly thought-provoking and imaginative paper. The experiments are creative and elegant and the data and analyses are, on the whole, very clear. Most importantly, the results provide considerable clarity to an emerging understanding of how longstanding ideas in psychology are at play in popular motor learning paradigms. One aspect of the paper that I remain slightly unconvinced of is the conceptual framing in terms of "working memory". The synthesis across psychology and motor learning is a considerable strength of the paper. However, it is still unclear to me in what sense both RC and MR are obviously instantiations of 'working memory'. In particular, RC is commonly conceived of in terms of 'model-free' learning. The authors appeal to Collins and Frank, who have argued that RL-like learning might reflect formation of working-memory-based representations, rather than necessarily corticostriatal learning mechanisms. but couldn't, in principle, the learning seen here be more corticostriatal-based than working-memory-based? The working memory narrative in the title feels a bit hasty/speculative...

We thank the reviewer for their comments on the framing of the paper, and we agree that the conceptual framing may appear speculative. However, in terms of mental rotation, previous psychophysical work has established it as a function of visual working memory (Hyun & Luck 2007). In terms of "response caching," we agree that this can be achieved both by a working memory look-up table and corticostriatal RL. We believe our analysis on "swap errors" (Figure 3) is more consistent with a working memory interpretation. However, we also believe it is likely that after practice (especially on longer training time scales than used here) the look-up table would "shift" to the RL system. We agree that we can make the discussion of the RL-like aspects more robust, and have done so in the Discussion. For these reasons, we would like to emphasize working memory in the title as we think it opens up a new avenue for understanding the mechanisms underlying different forms of cognitive strategies in sensorimotor learning, but we are open to deemphasizing working memory in the title if the reviewer requests we do so.

The discussion ends with the statement (line 577) that "working memory may operate on internally generated movement plans much like it operates on any number of sensory and abstract representations". In what sense does working memory "act" on anything? Isn't it a representation that can be acted upon, by something that can also act upon movement plans? The connection between working memory and motor learning will be something many in the motor learning field have not thought about much and this paper has the potential to be very didactic in this regard, so being crystal clear about these kind of statements seems essential.

We thank the reviewer for thinking our paper may be very didactic, and for underscoring the importance of using proper language here. We agree that our phrasing here was confusing and have revised that sentence as follows:

“Movement plans may be held and manipulated in working memory, perhaps in a manner consistent with visual, auditory, and tactile working memory. The results presented here suggest that spatial cognition plays a key role in motor learning.”

The 'interference effect' analysis (Figure 3B) is weak. The effect seems clear for the 2T groups. However, the comparison with the 12T group is inadequate. First, simply claiming a difference between groups based on demonstrating a significant effect for one group, but no significant effect for the other group, is flawed logic. There should instead be a group-by-delay interaction. However, the interaction comparison can't be done because there is no 1-back data point for the 12T group. That itself is a problem since there might be something privileged about 1-back trials that is driving the decline in compensation in the 2T groups and would also have enabled good performance in the 12T group, had there been any/enough 1-back trials. Also, if the method of compensation is simple response-caching, it's not clear that interference should necessarily manifest itself as a reversion towards 0 compensation. Inspecting distributions of errors might help clarify things, but overall I am skeptical that this analysis tells us anything about differing strategies of compensation in the two groups.

The reviewer makes a good point here. We were following the convention used by Collins and Frank, which has subtle differences in delay between the set sizes. However, we agree that our experimental design here is not well suited to an interpretable *post hoc* analysis of decay effects, especially as a group comparison. Thus, we have removed this analysis/figure-panel from the manuscript.

The Results contains quite a lot of methodological details. Given that a lot of quite different experiments are described, this makes for quite a lengthy Results section. The authors might consider streamlining the results a bit by describing only the most crucial aspects of the experimental design and providing pointers to other sections for the more specific details (e.g. lines 277-289 reads like it belongs in the methods section).

We thank the reviewer for their comment here. In the revised manuscript we have tried to limit the methodological details in the Results section to only the most necessary items.

Fig. 3A - "probability" doesn't quite seem the right label for the y-axis. I think it is really the probability of that error size given that an error occurred, not the overall probability of that error size. Perhaps "proportion of errors" would pose less risk of being misleading?

We agree and have edited the axis label.

512 - hyphenation: 12-target forced-reaction-time task

We have fixed this phrase.

Reviewer #2 (Remarks to the Author):

It is now well established that adaptation of reaching movements to a visuomotor rotation (i.e., rotation of a cursor representing the hand about the movement start point) involves two components: an explicit component, which can result in fast error reduction, and an implicit component, which is more gradual.

There is now a substantial amount of evidence that the explicit, or strategic, component involves a form of mental rotation whereby the participant reaches toward a location, or 'aim point', that is rotated away from the viewed target. Consistent with this interpretation, a number of previous studies have shown that this re-aiming strategy takes time to implement and that this time (referred to as reaction time or preparation time) appears to scale with the magnitude of the rotation that is implemented.

We enthusiastically agree with the reviewer on these two points. However, we would like to note that several years ago, the attribution of a large percentage of the visuomotor learning curve to explicit processes would have been heresy! We believe papers like Taylor et al., (2014), as well as many others now, have helped establish this fact. Importantly, however, these findings have largely remained descriptive, providing evidence that explicit strategies are at play, but not addressing the cognitive mechanisms underlying these strategies. We think the most convincing previous clues for a possible connection between explicit visuomotor learning and mental rotation comes from Fernandez-Ruiz et al (2011); we believe part of the advance made in our study is that we have made concrete some of the hypotheses laid out in that paper. While implicit learning in visuomotor adaptation tasks has a strong theoretical framework (i.e. internal models and sensory prediction errors), explicit learning is still poorly understood. Here we have provided strong, direct evidence describing underlying algorithms of explicit motor learning. Moreover, we also believe that the ideas presented in this manuscript have not been widely considered in the field, and thus our paper can have a strong impact in this regard.

This paper is very clearly written and easy to read. I have no concerns about the methods and results. All three experiments are very well designed, and all of the analyses are very well done. I especially liked the analysis of errors. For the introduction, I would have preferred a more direct consideration of the visuomotor task, leaving consideration of other tasks (i.e., the piano analogy which I was not fully convinced by) for the discussion. However, this is a matter of taste and I would not insist on any changes.

We are pleased to hear that the paper was easy to read and our experiment and analyses were done. We have revised parts of the Introduction, though at the moment have kept the piano analogy because we thought it was the most easily accessible example that conveys the two "bookend" forms of strategies people can employ, as well as the touching upon the ultimate goal of proceduralizing the S-R mapping. However, we would be willing to remove it at the reviewer's request.

Although I really liked this paper, I have to admit that I did not find the results particularly surprising and it seems to me that the basic results are predicted by previous work (see below). Perhaps I am being unfairly harsh here and would welcome a rebuttal by the authors.

The idea that participants learn stimulus-response pairings when only presented with targets are two locations (separated by 180°) does not seem very surprising. Indeed, a similar effect has been shown for mental rotation, per se. Thus, Provost and colleagues (2013) have shown that, after extensive practice with a small set of stimuli, mental rotation becomes very fast and no longer depends on rotation angle. In

contrast, with a larger set of stimuli, mental rotation becomes faster but continues to depend on rotation angle. Provost and colleagues concluded that “extensive practice with a small set of stimuli allowed participants to switch from a mental rotation strategy...to direct retrieval from memory of the correct response associated with each stimulus”. They also note that “the development of a retrieval strategy is generally consistent with Logan’s (1988) instance theory of automaticity.”

The authors cite the Provost et al. paper and do indicate that there is considerable overlap with their results. I guess I feel that the overlap is substantial that the current results are predictable from these previous results. (Note that the authors imply that the Provost et al. paper deals with “perceptual learning” but I am not convinced this is accurate.)

We thank the reviewer for their thoughtful comments and absolutely agree that the results here (in experiment 1 primarily) are indeed consistent with previous work we cited (Provost et al. 2013 in particular). We note, however, that the results from Provost et al. 2013 only consider mental rotation in and of itself. Here, we expanded their concepts into sensorimotor learning, where only a few studies have even considered the idea of mental rotation in the planning of movements (e.g., Fernandez et al. 2011, Georgopoulos & Massey, 1987). As such, we raise several points below that emphasize why we believe our study is novel and will have a high impact in the field, and also be of general interest. We note that point three highlights a new experiment we conducted and added to the manuscript. This experiment shows how strategies adopted during learning appear to shape motor flexibility and generalization. We would also like to thank the reviewer for inviting a rebuttal.

- 1) **Evidence for “motor” working memory:** Classic mental rotation (of visual percepts) has been characterized as an instantiation of visual working memory. In a nice study by Hyun & Luck (2007), the authors showed that adding load to the visual working memory system, but not the spatial working memory system, interfered with visual mental rotation performance. However, in terms of mental rotation of an imagined movement plan, the relevant working memory system is unknown. Previous research from Georgopoulos and colleagues suggests that a) motor mental rotation is correlated with visual mental rotation (in terms of rotation speed) within-subject, and b) neurons in primary motor cortex appear to reflect this mental rotation process (at least when formalized as the rotation of a 2D population vector). We believe it is the analog computation on the *motor response* (e.g. a ballistic arm movement) that makes our results a novel advance relative to the Provost et al. study. Without our experiments here, it’s not clear if parametric strategies are a specific kind of computation for mental imagery of visual objects, or a more “global” kind of computation that can bridge perception and action. Indeed, we do not think it’s a forgone conclusion that improving on a visuospatial skill like classic mental rotation would necessarily rely on the same mechanisms as visuomotor learning. Indeed, parametric versus discrete strategies could reflect a nice general framework for skill learning writ large, including *both* visuospatial and motor skills. Logan’s (1988) theory proposed a shift to discrete retrieval like we see here; we think the “parametric” stage we elucidate in our manuscript sharpens and extends this idea, and provides empirical support to the theory’s relevance to motor learning, which has not been fully explored in the motor learning literature where the primary focus has been on implicit adaptation processes associated with learning a forward model. Lastly, a more philosophical rebuttal here is that while we think our results are novel, they also have a very solid empirical and theoretical foundation, perhaps making them not hugely surprising once everything has been clearly

linked (which we believe our manuscript does). We think this makes our results particularly replicable and reliable, and hope this will be considered alongside novelty.

- 2) **Within-trial revelation of discrete vs. parametric strategies:** We agree that experiment 1 in our manuscript could be viewed as an extension of previous findings (Provost et al., 2013) to motor learning. However, we think the other three experiments in our manuscript (experiments 2-4) represent clear scientific advances. Using a forced-response-time task, we were able to reveal the algorithmic nature of the proposed “parametric” and “discrete” algorithms. To our knowledge, “intermediate” states of visual mental rotation have never been directly shown in continuous motor behavior (the closest attempt involves indirect measures of RT, Cooper, 1976). In experiment 2 we showed how under a parametric strategy, movement angles proceed through predictable intermediate states, which aligns with previous neurophysiology. Furthermore, in experiment 3 we showed how a under a discrete strategy, discrete distributions of responses are elicited, which directly supports the retrieval of cached responses. While these cognitive representations are implied by the RT results from the Provost et al. paper, we believe we have made a novel, deeper advance by actually “reading out” dynamic cognitive operations in motor behavior. In this vein, we are also making a closer connection between the behavior and the putative underlying neural mechanisms (whether it’s true population vector rotation or a form of motor averaging).
- 3) **The role of different strategies in flexibility and generalization:** We performed an additional experiment that investigated the downstream effects of different cognitive motor learning strategies. While transfer studies of mental rotation to unseen shapes have been conducted before (including by Provost et al. in one of their conditions), here we show how people extrapolate a parametric strategy in a spatial dimension, rather than across discrete visual objects. As described below, when trained under a high set size regime (which should bias subjects towards a parametric strategy), subjects show wide generalization to new regions of space, suggesting that they have learned the “structure” of the task and can thus extrapolate the rule. However, when trained under a low set size regime (which may lead to a more discrete strategy), subjects show graded generalization, suggesting a more “local” form of learning. We believe that our new study is a fundamental advance, showing how working memory representations during skill learning have downstream effects on the flexibility of that skill. Furthermore, generalization of visuomotor adaptation has mostly been discussed in terms of the *implicit* component of learning; here we provide a novel mechanistic hypothesis for how the *explicit* component should generalize depending on the particular strategy adopted. While previous work has assayed the generalization pattern for the explicit learning component (Heuer and Hegele 2010; McDougle et al., 2017), these previous studies have not addressed why or how the particular pattern of explicit generalization arises. This result also has many implications for previously-described generalization functions that were likely contaminated by explicit learning, and also show how the specific learning environment (e.g. the number of targets) plays a crucial role in generalization. There is a lot of exciting future research on this topic to be done and we believe we are just scratching the surface.

Below we summarize the method and results for the new extrapolation/generalization experiment (experiment 4), which has been added to the manuscript:

Subjects were trained on either 2 targets (2T) or 8 targets (8T) for 144 trials, with 50% feedback and 50% no feedback (to get them accustomed to no-FB trials, which would be later employed to assay the generalization function), and delayed cursor feedback on feedback trials (to extinguish

implicit learning, similar to the other experiments). Then, in a generalization probe, subjects were told to move to new probe targets (with no feedback) in whichever manner they thought would be successful if they were to see feedback. Critically, the generalization probes required extrapolation of learning from the training region to the new region. Target positions and rotation signs were counterbalanced across subjects.

As shown below, subjects in the 8T group showed a more global generalization pattern, while subjects in the 2T group showed more local generalization. We took a regression approach to quantify this effect, as shown in the third panel. [Note: we also performed the standard Gaussian-fitting analysis, as shown in another new figure, Fig. S6. However, many subjects, especially in the 8T condition, showed an unstable trade-off during fitting between the free parameters due to virtually complete generalization. While the results of the Gaussian fit were comparable to the regression (significantly larger width in the 8T group; Fig. S6), the linear regression approach was more interpretable.]

As predicted, the angular distance of probe targets from the nearest trained targets negatively affected generalization in the 2T group relative to the 12T group ($p = 0.025$). Specifically, under a parametric regime, subjects' patterns of generalization show signatures of "structure" learning. Conversely, under a discrete S-R regime, subjects' patterns of generalization show more localized "instance" learning, as predicted by experiments 1-3. We think this new experiment provides a

strong, novel test of the downstream consequences of using different strategies during motor learning and thus expands the scope of the manuscript.

The current results also seem similar to a paper on biorxiv: Huberdeau, Krakauer and Haith, “Practice induces a qualitative change in the memory representation for visuomotor learning”, <http://dx.doi.org/10.1101/226415>). Huberdeau and colleagues examined adaptation to a visuomotor rotation, manipulated preparation time, and used a small number of target locations (but did not contrast this with a large number of target locations as in the current study). Huberdeau et al. found that with extensive practice learning, or relearning, to reach under visuomotor rotations, the preparation time required to implement the strategic component (initially involving mental rotation) becomes shorter and shorter, to the point at which it can reasonably be referred to as ‘automatic’. I am not sure of the status of papers on biorxiv. However, it seems to me that this paper should be discussed.

We completely agree with the reviewer on this point. Indeed, we think the Huberdeau et al. study is a very nice counterpart to our study, and they fit together rather well. Our study attempts to characterize these strategic components in terms of specific psychological (and perhaps neural) mechanisms, and the Huberdeau study shows how strategies may become proceduralized in the long term. We have added a citation and discussion of this preprint.

ADDITIONAL COMMENTS

My only minor comment concerns the use of the term “bootstrapping the learning curve”, which I found very odd and not appropriate.

We agree and have removed this analogy.

Reviewer #3 (Remarks to the Author):

Authors make a strong case in support of the involvement of two working memory strategies during the reaction to visuomotor perturbations; stimulus response caching and mental rotation. These results are interesting and novel for the case of visuomotor explicit adaptation. However, the involvement of stimulus response and algorithm driven responses in control and learning has been proposed before in the discussion about improvement in mental rotation (that could be explained by a transition to an automatic response (Provost et al., 2012, Tarr & Pinker, 1989), discussed by authors), with the mental line results that the authors present, and dealt with in the context of the instance theory of automaticity by Logan (1988). In fact, it is interesting to compare these results with the model of Logan, especially the timed response results that shed light on the underlying running processes.

We thank the reviewer for their comments and strongly agree with their insightful interpretation. As mentioned in our response to Reviewer #2, we have broadened our discussion of the Provost et al. results. We also believe that our results do provide multiple important novel extensions of previous ideas. First, our results extend the findings of Provost et al. from visuospatial skills to motor skills. We think this is a critical and novel extension — joining these literatures is very important for the study of general learning principles. Second, we agree that our findings on the constrained response time tasks verify some of Logan’s (1988) ideas and believe we have provided novel support for it here. Third, as discussed above, we have added a new experiment (experiment 4) that reveals how different working memory strategies lead to different patterns of generalization. We think this new study is an important advance for understanding the role of

strategies in shaping motor learning, affording flexibility and transfer, and it also provides a novel mechanistic framework for understanding how generalization in motor learning is affected by the specific cognitive strategy the learner adopts.

I assume that the algorithms involved in motor learning may vary as a function of task and therefore not sure whether this is a general model for motor learning or a specific model for visuomotor rotation.

We thank the reviewer for this comment and think it is a very important point. We opted to focus on rotations for two main reasons: 1) They allow us to connect ideas and findings from two large literatures, one on classic mental rotation and one on visuomotor rotation learning. 2) Directional data is well-suited to the forced-response time task, as it is a continuous spatial measurement that can be (mostly) separated from movement extent. Critically, it has been shown that parametric RT effects also occur in the computation of both *gains* and *rotations* (Bhat & Sanes, 1998), suggesting that parametric strategies can be applied to movement extent and/or direction. Indeed, in that paper there was a correlation between subjects' rotation and gain computation rates. For our purposes, elucidating parametric gain computations in a forced-response time task would be experimentally tricky given the inertia of the limb. We believe the aforementioned results on linear RT effects in gain tasks are strongly suggestive of general-purpose parametric computations, and have added a further discussion of this important issue to the Discussion section.

Specific points:

Page 6 115 – please indicate that this is a between-subject design

We have added this to the manuscript.

Page 7 – reaction time is likely to also change as a function of error and success...12T 75deg during the late phase show increased reaction time and increased errors. lack of success may indicate increased cautiousness rather than a qualitative difference in strategy.

We agree that, in theory, a kind of speed-accuracy trade off could affect our results. However, the results of our ANOVA did not reveal significant differences in error. Moreover, as shown in Figure 2B, the 12T 75° group appeared to be fully at asymptote by the end of learning. Thus, we do not believe error rates drove our RT results. As shown in Figure 9, mental rotation predictions from an out-of-set sample of subjects (the group in exp. 2) predicted RT differences in the subjects of experiment 1, making mental rotation a more parsimonious explanation.

Page 7 - The Lack of consistency with Hick's law should be discussed

This is a subtle and important point and we thank the reviewer for bringing it up. In fact, the inconsistency with Hick's Law is a key point in experiment 1 — in this task, "stimulus" uncertainty (where will the target appear, i.e. set size) interacts with both rotation size and the degree of learning. Thus, when a "global" strategy for stimulus-response learning (e.g. mental rotation) is adopted, the RT costs predicted by Hick's Law can be averted. We have added a discussion of this important point to the manuscript.

Page 8 178 – are there reference in support of reduced implicit adaptation as a function of delay? It could

be that the increased delay between response and feedback also affected the estimation of the after effects and led to an under-estimation of the effect.

We thank the reviewer for their comment on this. Indeed, there are multiple lines of evidence for the effect of delay on implicit adaptation (Kitazawa et al., 1995; Brudner et al., 2016; Schween et al., 2017; Parvin et al., 2018). Consistent with these studies, while aftereffects are normally in the range of 10°-20°, ours were, on average, all < 5°.

Experiment 1 aims both to validate the overall idea of two strategies in explicit adaptation and to examine specific questions about the involvement of the mechanisms in certain cases (2t 75 and 12t 25). This logic seems a bit circular. A clear prediction about the two conditions in the middle may be helpful in this case.

We thank the reviewer for this request. In essence, our primary prediction was a three-way interaction between practice, rotation size, and set size on RT, due to a time-based trade-off between a parametric strategy (early) and a discrete strategy (late). Thus, the middle conditions are necessary to test for the interactions. We have made this more explicit in the manuscript.

Page 9 202 the swap errors make sense, but the flip errors not. Is there any support for this prediction? Can authors provide additional intuition for this prediction?

We thank the reviewer for their comment, this is an important question. First, a recent study (Christou et al., 2016) showed clear evidence of explicit learning in a rotation task going in the wrong direction but with the correct magnitude. Moreover, variable aiming reports during the initial stages of learning in individuals often shows large swings in the sign of the hand direction that cannot be explained by traditional implicit motor adaptation accounts (Taylor et al., 2014). Thus, we intuited that large errors would be, on average, sign flips.

Page 10 215 – can this effect explained by a time difference between the presentations (and therefore reflect a passive decay process)? Or alternatively by a switching effect between the targets?

This is a good point and we think the answer is not clear given our results. Because of this and some other good points made by Reviewer #1, we have actually removed the decay analysis as it is difficult to interpret given our specific experimental design.

Page 12 – in exp 2 feedback was delayed by 500ms and not by 2s as before. This difference should be mentioned and discussed.

This decision was made to a) maximize the number of trials in experiment 2 given the noise of forced-rt data, b) because previous research suggests that 500ms should be enough to limit aftereffects (Kitazawa et al., 1995), and c) because large rotations appear to show diminished implicit learning (Morehead et al. 2017). Indeed, aftereffects in experiment 2 were actually *smaller* than those in experiment 1, even though the delay was briefer. We have added a justification of this method difference to the Methods section.

Page 12 268 – study design assumes full generalization between perturbation target and subsequent aiming target. Could narrower generalization provide an alternative explanation to the results?

This is a critical point and we believe our new experiment (experiment 4, see response to Reviewer #2 above) directly addresses this issue.

Reviewers' Comments:

Reviewer #1:

None

Reviewer #2:

Remarks to the Author:

As I indicated in my original review, I felt that this paper was well written, that the experiments were well designed, and that the analyses were appropriate and well executed.

My main concern was about novelty and whether the results were really all that surprising given (a) previous work in motor learning suggesting that mental rotation underlies the explicit component of adaptation to a visuomotor rotation, and (b) previous work on mental rotation tasks showing that mental rotation can become automatized.

I was somewhat on the fence about this concern because (a) I recognize that 'mental rotation' of a reach direction may differ, at least in some respects, to mental rotation of a visual stimulus, and (b) the motor learning field may not be generally aware of the connection between these two bodies of work.

The authors have made these points (and others) in their rebuttal. They have also added a new experiment that further distinguishes the two forms of learning they find. The new experiment is a nice addition to the paper. However, it is the points the authors have made that have swayed me to be positive about this paper. I appreciate the fact that the authors directly acknowledge the relevance of previous work while arguing that their work represents an advance.

My other concern related to a paper by Huberdeau and colleagues, on biorxiv, that is in many ways similar to the current work. The authors cite this work in the revised manuscript and I accept that the papers are complementary.

I have not specific comments for further revision.

Reviewer #3:

Remarks to the Author:

The authors addressed most of the concerns that were raised by the reviewers and convinced me that the series of studies contribute to our understanding of explicit strategies during sensorimotor adaptation learning.

authors also added another experiment in order to further differentiate between the two suggested mechanisms by describing their generalization properties. while the presented results support the existence of a difference between the two training conditions, it is still not clear why the difference in generalization patterns should go in that direction. furthermore, it could be argued that the difference in generalization is a result of the difference in the training condition (variability of reaches, number of targets etc.) rather than the strategy that subjects chose to counter the perturbations.

Perhaps the data from the free condition in experiment 2, where subjects learned how to react to a perturbation that was presented to a different target (if i followed the methods, the target changed between the two reaching movements within a trial), could add substance to their hypothesis.

We thank the reviewers for their thoughtful comments. Reviewer #1 and Reviewer #2 did not request any further comments or modifications to the text.

REVIEWER #3

Authors also added another experiment in order to further differentiate between the two suggested mechanisms by describing their generalization properties. While the presented results support the existence of a difference between the two training conditions, it is still not clear why the difference in generalization patterns should go in that direction.

Under traditional views of response caching (or relatedly, model-free reinforcement learning), the response should be specific to the conditions where the association was learned. As such, as conditions begin to deviate from the learned stimulus, then the associated response would be less likely to be triggered. This would be reflected as a relatively narrow generalization function: Subject's learned response should fall off as a function of the angular distance between the trained (learned) target location and the new (unlearned) target location. In contrast, there is no *a priori* reason to believe that a parametric strategy would be limited based on the training conditions. Indeed, as *Experiments 1* and *2* demonstrated, it is in fact the variability in training (i.e., the number of stimulus-response associations that need to be learned) that guides subject's to a more flexible, algorithmic strategy.

This reasoning has been highlighted in the manuscript on lines 1376-1379:

"We reasoned that subjects using a discrete response caching strategy (RC) would show diminished generalization relative to subjects using a parametric mental rotation strategy (MR). This could occur because under an RC regime, specific local instances are learned, whereas under an MR regime, a global rule (or structure) is learned that can be applied indiscriminately."

Furthermore, it could be argued that the difference in generalization is a result of the difference in the training condition (variability of reaches, number of targets etc.) rather than the strategy that subjects chose to counter the perturbations. Perhaps the data from the free condition in experiment 2, where subjects learned how to react to a perturbation that was presented to a different target (if i followed the methods, the target changed between the two reaching movements within a trial), could add substance to their hypothesis.

We sympathize with the reviewer's point that the differences in generalization could be the result of several variables, such as the variability of reaches, number of targets, etc. Based on the findings from experiments 1-3, it is particularly the inherent differences between the number of targets that proved critical in determining the type of strategy used. This is most strongly supported by the FORCED task results from *Experiments 2* and *3*, which directly assayed the type of strategies employed. Thus, as the reviewer points out, it is necessary to lean on the findings from the previous experiments to provide a more mechanistic explanation for the difference in generalization functions between the 2T and 8T conditions in *Experiment 4*, which we explicitly state in the Results section (lines 1384-1385):

"While this experiment could not directly infer subjects' learning strategies as in Experiments 1-3, we reasoned that the set size manipulation would bias subjects toward either RC (2T) or MR (8T)."